# Sparse Deep Learning: A New Framework Immune to Local Traps and Miscalibration

**Yan Sun**
Purdue University
West Lafayette, IN 47906
sun748@purdue.edu

**Wenjun Xiong**
Guangxi Normal University & Purdue University
West Lafayette, IN 47906
xiong90@purdue.edu

**Faming Liang**[*]
Purdue University
West Lafayette, IN 47906
fmliang@purdue.edu

## Abstract

Deep learning has powered recent successes of artificial intelligence (AI). However, the deep neural network, as the basic model of deep learning, has suffered from issues such as local traps and miscalibration. In this paper, we provide a new framework for sparse deep learning, which has the above issues addressed in a coherent way. In particular, we lay down a theoretical foundation for sparse deep learning and propose prior annealing algorithms for learning sparse neural networks. The former has successfully tamed the sparse deep neural network into the framework of statistical modeling, enabling prediction uncertainty correctly quantified. The latter can be asymptotically guaranteed to converge to the global optimum, enabling the validity of the down-stream statistical inference. Numerical result indicates the superiority of the proposed method compared to the existing ones.

**Keywords**: Asymptotic Normality, Posterior Consistency, Prior Annealing, Structure Selection, Uncertainty Quantification

## 1 Introduction

During the past decade, deep neural networks (DNNs) have achieved the state-of-the-art performance in many machine learning tasks such as computer vision and natural language processing. However, the DNN suffers from a training-prediction dilemma from the perspective of statistical inference: *A small DNN model can be well calibrated, but tends to get trapped into a local optimum; on the other hand, an over-parameterized DNN model can be easily trained to a global optimum (with zero training loss), but tends to be miscalibrated [14].* In consequence, it is often unclear whether a DNN is guaranteed to have a desired property after training instead of getting trapped into an arbitrarily poor local minimum, or whether its decision/prediction is reliable. This difficulty makes the trustworthiness of AI highly questionable.

To resolve this difficulty, researchers have attempted from two sides of the training-prediction dilemma. Towards understanding the optimization process of the DNN training, a line of researches have been done. For example, [13] and [27] studied the training loss surface of over-parameterized DNNs. They showed that for a fully connected DNN, almost all local minima are globally optimal, if the width of one layer of the DNN is no smaller than the training sample size and the network

---

[*]To whom correspondence should be addressed: Faming Liang

35th Conference on Neural Information Processing Systems (NeurIPS 2021).

structure from this layer on is pyramidal. Recently, [1, 8, 35] and [36] explored the convergence theory of the gradient-based algorithms in training over-parameterized DNNs. They showed that the gradient-based algorithms with random initialization can converge to global minima provided that the width of the DNN is polynomial in training sample size.

To improve calibration of the DNN, different methods have been developed, see e.g., Monte Carlo dropout [12] and deep ensemble [18]. However, these methods did not provide a rigorous study for the asymptotic distribution of the DNN prediction and thus could not correctly quantify its uncertainty. Recently, researchers have attempted to address this issue with sparse deep learning. For example, for Bayesian sparse neural networks, [19], [28] and [31] established the posterior consistency, and [33] further established the Bernstein-von Mises (BvM) theorem for linear and quadratic functionals. The latter guarantees in theory that the Bayesian credible region has a faithful frequentist coverage. However, since the theory by [33] does not cover the point evaluation functional, the uncertainty of the DNN prediction still cannot be correctly quantified. Moreover, their theory is developed with the spike-and-slab prior (i.e., each weight or bias of the DNN is subject to a spike-and-slab prior), whose discrete nature makes the resulting posterior distribution extremely hard to simulate. To facilitate computation, [31] employed a mixture Gaussian prior. However, due to nonconvexity of the loss function of the DNN, a direct MCMC simulation still cannot be guaranteed to converge to the right posterior distribution even with the mixture Gaussian prior.

In this paper, we provide a new framework for sparse deep learning, which successfully resolved the training-prediction dilemma. In particular, we propose two prior annealing algorithms, one from the frequentist perspective and one from the Bayesian perspective, for learning sparse neural networks. The algorithms start with an over-parameterized deep neural network and then have its structure gradually sparsified. We provide a theoretical guarantee that the training procedures are immune to local traps, the resulting sparse structures are consistent, and the predicted values are asymptotically normally distributed. The latter enables the prediction uncertainty correctly quantified. Our contribution in this paper is two-fold:

- We provide a new framework for sparse deep learning, which is immune to local traps and miscalibration.

- We lay down the theoretical foundation for how to make statistical inference with sparse deep neural networks.

The remaining part of the paper is organized as follows. Section 2 lays down the theoretical foundation for sparse deep learning. Section 3 describes the proposed prior annealing algorithms. Section 4 presents some numerical results. Section 5 concludes the paper.

## 2   Theoretical Foundation for Sparse Deep Learning

As mentioned previously, sparse deep learning has received much attention as a promising way for addressing the miscalibration issue of the DNN. Theoretically, the approximation power of the sparse DNN has been studied for various classes of functions [30, 3]. Under the Bayesian setting, posterior consistency has been established in [19, 28, 31]. In particular, the work [31] has achieved important progress toward taming sparse DNNs into the framework of statistical modeling. They provide a neural network approximation theory fundamentally different from the existing ones. In the existing theory, no data is involved and a small network can potentially achieve an arbitrarily small approximation error by allowing connection weights to take values in an unbounded space[24]. In contrast, the theory by [31] links the network approximation error, the network size, and the bound of connection weights to the training sample size. They prove that for a given training sample size $n$, a sparse DNN of size $O(n/\log(n))$ has been large enough to approximate many types of functions, such as affine functions and piecewise smooth functions, arbitrarily well as $n \to \infty$. Moreover, they prove that the sparse DNN possesses many theoretical guarantees. For example, its structure is more interpretable, from which the relevant variables can be consistently identified for high-dimensional nonlinear systems; and its generalization error bound is asymptotically optimal.

From the perspective of statistical inference, some gaps remain toward taming sparse DNNs into the framework of statistical modeling. This paper bridges the gap by establishing (i) asymptotic normality of the connection weights, and (ii) asymptotic normality of the prediction.

## 2.1 Posterior consistency and structure selection consistency

This subsection provides a brief review of the sparse DNN theory developed in [31] and gives the conditions that we will use in the followed theoretical developments. Without loss of generality, we let $D_n = (\boldsymbol{x}^{(i)}, y^{(i)})_{i=1,\ldots,n}$ denote a dataset of $n$ *i.i.d* observations, where $\boldsymbol{x}^{(i)} \in R^{p_n}$ and $y^{(i)} \in R$. Consider a generalized linear model with the distribution of $y$ given by

$$f(y|\mu^*(\boldsymbol{x})) = \exp\{A(\mu^*(\boldsymbol{x}))y + B(\mu^*(\boldsymbol{x})) + C(y)\},$$

where $\mu^*(\boldsymbol{x})$ is a nonlinear function of $\boldsymbol{x}$ and $A(\cdot)$, $B(\cdot)$ and $C(\cdot)$ are appropriately defined functions. For example, for normal regression, we have $A(\mu^*) = \mu^*/\sigma^2$, $B(\mu^*) = -\mu^{*2}/2\sigma^2$, $C(y) = -y^2/2\sigma^2 - \log(2\pi\sigma^2)/2$, and $\sigma^2$ is a constant. We approximate $\mu^*(\boldsymbol{x})$ using a fully connected DNN with $H_n - 1$ hidden layers. Let $L_h$ denote the number of hidden units at layer $h$ with $L_{H_n} = 1$ for the output layer and $L_0 = p_n$ for the input layer. Let $\boldsymbol{w}^h \in \mathbb{R}^{L_h \times L_{h-1}}$ and $\boldsymbol{b}^h \in \mathbb{R}^{L_h \times 1}$, $h \in \{1, 2, \ldots, H_n\}$ denote the weights and bias of layer $h$, and let $\psi^h : \mathbb{R}^{L_h \times 1} \to \mathbb{R}^{L_h \times 1}$ denote a coordinate-wise and piecewise differentiable activation function of layer $h$. The DNN forms a nonlinear mapping

$$\mu(\boldsymbol{\beta}, \boldsymbol{x}) = \boldsymbol{w}^{H_n} \psi^{H_n-1} \left[ \cdots \psi^1 \left[ \boldsymbol{w}^1 \boldsymbol{x} + \boldsymbol{b}^1 \right] \cdots \right] + \boldsymbol{b}^{H_n}, \tag{1}$$

where $\boldsymbol{\beta} = (\boldsymbol{w}, \boldsymbol{b}) = \{\boldsymbol{w}_{ij}^h, \boldsymbol{b}_k^h : h \in \{1, \ldots, H_n\}, i, k \in \{1, \ldots, L_h\}, j \in \{1, \ldots, L_{h-1}\}\}$ denotes the collection of all weights and biases, consisting of $K_n = \sum_{h=1}^{H_n} (L_{h-1} \times L_h + L_h)$ elements. For convenience, we treat bias as a special connection and call each element in $\boldsymbol{\beta}$ a connection weight. In order to represent the structure for a sparse DNN, we introduce an indicator variable for each connection weight. Let $\boldsymbol{\gamma}^{\boldsymbol{w}^h}$ and $\boldsymbol{\gamma}^{\boldsymbol{b}^h}$ denote the indicator variables associated with $\boldsymbol{w}^h$ and $\boldsymbol{b}^h$, respectively. Let $\boldsymbol{\gamma} = \{\boldsymbol{\gamma}_{ij}^{\boldsymbol{w}^h}, \boldsymbol{\gamma}_k^{\boldsymbol{b}^h} : h \in \{1, \ldots, H_n\}, i, k \in \{1, \ldots, L_h\}, j \in \{1, \ldots, L_{h-1}\}\}$, which specifies the structure of the sparse DNN. With slight abuse of notation, we will also write $\mu(\boldsymbol{\beta}, \boldsymbol{x})$ as $\mu(\boldsymbol{\beta}, \boldsymbol{x}, \boldsymbol{\gamma})$ to include the information of the network structure. We assume $\mu^*(\boldsymbol{x})$ can be well approximated by a *parsimonious neural network* with relevant variables, and call this parsimonious network as the *true DNN model*. More precisely, we define the *true DNN model* as

$$(\boldsymbol{\beta}^*, \boldsymbol{\gamma}^*) = \underset{(\boldsymbol{\beta}, \boldsymbol{\gamma}) \in \mathcal{G}_n, \, \|\mu(\boldsymbol{\beta}, \boldsymbol{\gamma}, \boldsymbol{x}) - \mu^*(\boldsymbol{x})\|_{L^2(\Omega)} \leq \varpi_n}{\arg\min} |\boldsymbol{\gamma}|, \tag{2}$$

where $\mathcal{G}_n := \mathcal{G}(C_0, C_1, \varepsilon, p_n, H_n, L_1, L_2, \ldots, L_{H_n})$ denotes the space of valid sparse networks satisfying condition A.2 (given below) for the given values of $H_n$, $p_n$, and $L_h$'s, and $\varpi_n$ is some sequence converging to 0 as $n \to \infty$. For any given DNN $(\boldsymbol{\beta}, \boldsymbol{\gamma})$, the error $\mu(\boldsymbol{\beta}, \boldsymbol{\gamma}, \boldsymbol{x}) - \mu^*(\boldsymbol{x})$ can be generally decomposed as the network approximation error $\mu(\boldsymbol{\beta}^*, \boldsymbol{\gamma}^*, \boldsymbol{x}) - \mu^*(\boldsymbol{x})$ and the network estimation error $\mu(\boldsymbol{\beta}, \boldsymbol{\gamma}, \boldsymbol{x}) - \mu(\boldsymbol{\beta}^*, \boldsymbol{\gamma}^*, \boldsymbol{x})$. The $L_2$ norm of the former is bounded by $\varpi_n$, and the order of the latter will be given in Lemma 2.1. For the sparse DNN, we make the following assumptions:

- A.1 The input $\boldsymbol{x}$ is bounded by 1 entry-wisely, i.e. $\boldsymbol{x} \in \Omega = [-1, 1]^{p_n}$, and the density of $\boldsymbol{x}$ is bounded in its support $\Omega$ uniformly with respect to $n$.

- A.2 The true sparse DNN model satisfies the following conditions:

  - A.2.1 The network structure satisfies: $r_n H_n \log n + r_n \log \overline{L} + s_n \log p_n \leq C_0 n^{1-\varepsilon}$, where $0 < \varepsilon < 1$ is a small constant, $r_n = |\boldsymbol{\gamma}^*|$ denotes the connectivity of $\boldsymbol{\gamma}^*$, $\overline{L} = \max_{1 \leq j \leq H_n-1} L_j$ denotes the maximum hidden layer width, $s_n$ denotes the input dimension of $\boldsymbol{\gamma}^*$.

  - A.2.2 The network weights are polynomially bounded: $\|\boldsymbol{\beta}^*\|_\infty \leq E_n$, where $E_n = n^{C_1}$ for some constant $C_1 > 0$.

- A.3 The activation function $\psi$ is Lipschitz continuous with a Lipschitz constant of 1.

Refer to [31] for explanations and discussions on these assumptions. We let each connection weight and bias be subject to a mixture Gaussian prior, i.e.,

$$w_{ij}^h \sim \lambda_n N(0, \sigma_{1,n}^2) + (1 - \lambda_n) N(0, \sigma_{0,n}^2), \quad b_k^h \sim \lambda_n N(0, \sigma_{1,n}^2) + (1 - \lambda_n) N(0, \sigma_{0,n}^2), \tag{3}$$

where $\lambda_n \in (0, 1)$ is the mixture proportion, $\sigma_{0,n}^2$ is typically set to a very small number, while $\sigma_{1,n}^2$ is relatively large.

**Posterior Consistency** Let $P^*$ and $E^*$ denote the respective probability measure and expectation with respect to data $D_n$. Let $d(p_1, p_2)$ denote the Hellinger distance between two densities $p_1(\boldsymbol{x}, y)$ and $p_2(\boldsymbol{x}, y)$. Let $\pi(A \mid D_n)$ be the posterior probability of an event $A$.

**Lemma 2.1.** *(Theorem 2.1 of [31]) Suppose Assumptions A.1-A.3 hold. If the mixture Gaussian prior (3) satisfies the conditions: $\lambda_n = O(1/\{K_n[n^{H_n}(\overline{L}p_n)]^\tau\})$ for some constant $\tau > 0$, $E_n/\{H_n \log n + \log \overline{L}\}^{1/2} \lesssim \sigma_{1,n} \lesssim n^\alpha$ for some constant $\alpha > 0$, and $\sigma_{0,n} \lesssim \min\{1/\{\sqrt{n}K_n(n^{3/2}\sigma_{1,0}/H_n)^{H_n}\}, 1/\{\sqrt{n}K_n(nE_n/H_n)^{H_n}\}\}$, then there exists an error sequence $\epsilon_n^2 = O(\varpi_n^2) + O(\zeta_n^2)$ such that $\lim_{n\to\infty} \epsilon_n = 0$ and $\lim_{n\to\infty} n\epsilon_n^2 = \infty$, and the posterior distribution satisfies*

$$
\begin{aligned}
P^* &\left\{\pi[d(p_{\boldsymbol{\beta}}, p_{\mu^*}) > 4\epsilon_n | D_n] \geq 2e^{-cn\epsilon_n^2}\right\} \leq 2e^{-cn\epsilon_n^2}, \\
E_{D_n}^* &\pi[d(p_{\boldsymbol{\beta}}, p_{\mu^*}) > 4\epsilon_n | D_n] \leq 4e^{-2cn\epsilon_n^2},
\end{aligned}
\tag{4}
$$

*for sufficiently large $n$, where $c$ denotes a constant, $\zeta_n^2 = [r_n H_n \log n + r_n \log \overline{L} + s_n \log p_n]/n$, $p_{\mu^*}$ denotes the underlying true data distribution, and $p_{\boldsymbol{\beta}}$ denotes the data distribution reconstructed by the Bayesian DNN based on its posterior samples.*

**Structure Selection Consistency** The DNN is generally nonidentifiable due to the symmetry of network structure. For example, $\mu(\boldsymbol{\beta}, \boldsymbol{\gamma}, \boldsymbol{x})$ can be invariant if one permutes certain hidden nodes or simultaneously changes the signs or scales of certain weights. As in [31], we define a set of DNNs by $\Theta$ such that any possible DNN can be represented by one and only one DNN in $\Theta$ via nodes permutation, sign changes, weight rescaling, etc. Let $\nu(\boldsymbol{\gamma}, \boldsymbol{\beta}) \in \Theta$ be an operator that maps any DNN to $\Theta$ via appropriate weight transformations. To serve the purpose of structure selection in $\Theta$, we consider the marginal inclusion posterior probability (MIPP) approach proposed in [20]. For each connection, we define its MIPP by $q_i = \int \sum_{\boldsymbol{\gamma}} e_{i|\nu(\boldsymbol{\gamma}, \boldsymbol{\beta})} \pi(\boldsymbol{\gamma}|\boldsymbol{\beta}, D_n)\pi(\boldsymbol{\beta}|D_n)d\boldsymbol{\beta}$ for $i = 1, 2, \ldots, K_n$, where $e_{i|\nu(\boldsymbol{\gamma}, \boldsymbol{\beta})}$ is the indicator of connection $i$. The MIPP approach is to choose the connections whose MIPPs are greater than a threshold $\hat{q}$, i.e., setting $\hat{\boldsymbol{\gamma}}_{\hat{q}} = \{i : q_i > \hat{q}, i = 1, 2, \ldots, K_n\}$ as an estimator of $\boldsymbol{\gamma}^* \in \Theta$. Let $A(\epsilon_n) = \{\boldsymbol{\beta} : d(p_{\boldsymbol{\beta}}, p_{\mu^*}) \geq \epsilon_n\}$ and define $\rho(\epsilon_n) = \max_{1 \leq i \leq K_n} \int_{A(\epsilon_n)^c} \sum_{\boldsymbol{\gamma}} |e_{i|\nu(\boldsymbol{\gamma}, \boldsymbol{\beta})} - e_{i|\nu(\boldsymbol{\gamma}^*, \boldsymbol{\beta}^*)}|\pi(\boldsymbol{\gamma}|\boldsymbol{\beta}, D_n)\pi(\boldsymbol{\beta}|D_n)d\boldsymbol{\beta}$, which measures the structure difference between the true and sampled models on $A(\epsilon_n)^c$. Then we have:

**Lemma 2.2.** *(Theorem 2.2 of [31]) If the conditions of Lemma 2.1 hold and $\rho(\epsilon_n) \to 0$ as $n \to \infty$ and $\epsilon_n \to 0$, then (i) $\max_{1 \leq i \leq K_n}\{|q_i - e_{i|\nu(\boldsymbol{\gamma}^*, \boldsymbol{\beta}^*)}|\} \xrightarrow{p} 0$; (ii) (sure screening) $P(\boldsymbol{\gamma}_* \subset \hat{\boldsymbol{\gamma}}_{\hat{q}}) \xrightarrow{p} 1$ for any pre-specified $\hat{q} \in (0, 1)$; (iii) (consistency) $P(\boldsymbol{\gamma}_* = \hat{\boldsymbol{\gamma}}_{0.5}) \xrightarrow{p} 1$.*

Lemma 2.2 implies consistency of variable selection for the true DNN model as defined in (2).

## 2.2 Asymptotic Normality of Connection Weights

In this section, we establish the asymptotic normality of the network parameters and predictions. Let $nl_n(\boldsymbol{\beta}) = \sum_{i=1}^n \log(p_{\boldsymbol{\beta}}(\boldsymbol{x}_i, y_i))$ denote the log-likelihood function, and let $\pi(\boldsymbol{\beta})$ denote the density of the mixture Gaussian prior (3). Let $h_{i_1, i_2, \ldots, i_d}(\boldsymbol{\beta})$ denote the $d$-th order partial derivatives $\frac{\partial^d l_n(\boldsymbol{\beta})}{\partial\beta_{i_1}\partial\beta_{i_2}\cdots\partial\beta_{i_d}}$. Let $H_n(\boldsymbol{\beta})$ denote the Hessian matrix of $l_n(\boldsymbol{\beta})$. Let $h_{ij}(\boldsymbol{\beta})$ and $h^{ij}(\boldsymbol{\beta})$ denote the $(i, j)$-th component of $H_n(\boldsymbol{\beta})$ and $H_n^{-1}(\boldsymbol{\beta})$, respectively. Let $\bar{\lambda}_n(\boldsymbol{\beta})$ and $\underline{\lambda}_n(\boldsymbol{\beta})$ denotes the maximum and minimum eigenvalue of the Hessian matrix $H_n(\boldsymbol{\beta})$, respectively. Let $B_{\lambda,n} = \bar{\lambda}_n^{1/2}(\boldsymbol{\beta}^*)/\underline{\lambda}_n(\boldsymbol{\beta}^*)$ and $b_{\lambda,n} = \sqrt{r_n/n}B_{\lambda,n}$, where $r_n$ is the connectivity of $\boldsymbol{\gamma}^*$. For a DNN parameterized by $\boldsymbol{\beta}$, we define the weight truncation at the true model $\boldsymbol{\gamma}^*$: $(\boldsymbol{\beta}_{\boldsymbol{\gamma}^*})_i = \boldsymbol{\beta}_i$ for $i \in \boldsymbol{\gamma}^*$ and $(\boldsymbol{\beta}_{\boldsymbol{\gamma}^*})_i = 0$ otherwise. For the mixture Gaussian prior (3), let $B_{\delta_n}(\boldsymbol{\beta}^*) = \{\boldsymbol{\beta} : |\boldsymbol{\beta}_i - \boldsymbol{\beta}_i^*| < \delta_n, \forall i \in \boldsymbol{\gamma}^*, |\boldsymbol{\beta}_i - \boldsymbol{\beta}_i^*| < 2\sigma_{0,n}\log(\frac{\sigma_{1,n}}{\lambda_n\sigma_{0,n}}), \forall i \notin \boldsymbol{\gamma}^*\}$. We follow the definition of asymptotic normality in [4] and [33]:

**Definition 2.1.** *Denote by $d_{\boldsymbol{\beta}}$ the bounded Lipschitz metric for weak convergence and by $\phi_n$ the mapping $\phi_n : \boldsymbol{\beta} \to \sqrt{n}(g(\boldsymbol{\beta}) - g_*)$. We say that the posterior distribution of the functional $g(\boldsymbol{\beta})$ is asymptotically normal with the center $g_*$ and variance $G$ if $d_{\boldsymbol{\beta}}(\pi[\cdot \mid D_n] \circ \phi_n^{-1}, N(0, G)) \to 0$ in $P^*$-probability as $n \to \infty$. We will write this more compactly as $\pi[\cdot \mid D_n] \circ \phi_n^{-1} \rightsquigarrow N(0, G)$.*

Theorem 2.1 establishes the asymptotic normality of $\tilde{\nu}(\boldsymbol{\beta})$, where $\tilde{\nu}(\boldsymbol{\beta})$ denotes a transformation of $\boldsymbol{\beta}$ which is invariant with respect to $\mu(\boldsymbol{\beta}, \boldsymbol{\gamma}, \boldsymbol{x})$ while minimizing $\|\tilde{\nu}(\boldsymbol{\beta}) - \boldsymbol{\beta}^*\|_\infty$.

**Theorem 2.1.** *Assume the conditions of Lemma 2.2 hold with $\rho(\epsilon_n) = o(\frac{1}{K_n})$ and $C_1 > \frac{2}{3}$ in Condition A.2.2. For some $\delta_n$ s.t. $\frac{r_n}{\sqrt{n}} \lesssim \delta_n \lesssim \frac{1}{\sqrt[3]{nr_n}}$, let $A(\epsilon_n, \delta_n) = \{\boldsymbol{\beta} : \max_{i \in \boldsymbol{\gamma}^*} |\boldsymbol{\beta}_i - \boldsymbol{\beta}_i^*| > \delta_n, d(p_{\boldsymbol{\beta}}, p_{\mu^*}) \leq \epsilon_n\}$, where $\epsilon_n$ is the posterior contraction rate as defined in Lemma 2.1. Assume there exists some constants $C > 2$ and $M > 0$ such that*

C.1 *$\boldsymbol{\beta}^* = (\boldsymbol{\beta}_1^*, \boldsymbol{\beta}_2^*, \ldots, \boldsymbol{\beta}_{K_n}^*)$ is generic [11, 10], $\min_{i \in \boldsymbol{\gamma}^*} |\boldsymbol{\beta}_i^*| > C\delta_n$ and $\pi(A(\epsilon_n, \delta_n) \mid D_n) \to 0$ as $n \to \infty$.*

C.2 *$|h_i(\boldsymbol{\beta}^*)| < M$, $|h_{j,k}(\boldsymbol{\beta}^*)| < M$, $|h^{j,k}(\boldsymbol{\beta}^*)| < M$, $|h_{i,j,k}(\boldsymbol{\beta})| < M$, $|h_l(\boldsymbol{\beta})| < M$ hold for any $i, j, k \in \boldsymbol{\gamma}^*$, $l \notin \boldsymbol{\gamma}^*$ and $\boldsymbol{\beta} \in B_{2\delta_n}(\boldsymbol{\beta}^*)$.*

C.3 *$\sup \left\{ |E_{\boldsymbol{\beta}}(a^T U)^3| : \|\boldsymbol{\beta}_{\boldsymbol{\gamma}^*} - \boldsymbol{\beta}^*\| \leq 1.2b_{\lambda,n}, \|a\| = 1 \right\} \leq 0.1\sqrt{n/r_n}\underline{\lambda}_n^2(\boldsymbol{\beta}^*)/\bar{\lambda}_n^{1/2}(\boldsymbol{\beta}^*)$ and $B_{\lambda,n} = O(1)$, where $U = Z - E_{\boldsymbol{\beta}_{\boldsymbol{\gamma}^*}}(Z)$, $Z$ denotes a random variable drawn from a neural network model parameterized by $\boldsymbol{\beta}_{\boldsymbol{\gamma}^*}$, and $E_{\boldsymbol{\beta}_{\boldsymbol{\gamma}^*}}(Z)$ denotes the mean of $Z$.*

*Then $\pi[\sqrt{n}(\tilde{\nu}(\boldsymbol{\beta}) - \boldsymbol{\beta}^*) \mid D_n] \rightsquigarrow N(0, \boldsymbol{V})$ in $P^*$-probability as $n \to \infty$, where $\boldsymbol{V} = (v_{ij})$, and $v_{i,j} = E(h^{i,j}(\boldsymbol{\beta}^*))$ if $i, j \in \boldsymbol{\gamma}^*$ and 0 otherwise.*

Condition C.1 is essentially an identifiability condition, i.e., when $n$ is sufficiently large, the DNN weights cannot be too far away from the true weights if the DNN produces approximately the same distribution as the true data. Condition C.2 gives typical conditions on derivatives of the DNN. Condition C.3 ensures consistency of the MLE of $\boldsymbol{\beta}^*$ for the given structure $\boldsymbol{\gamma}^*$ [29].

### 2.2.1 Asymptotic Normality of Prediction

Theorem 2.2 establishes asymptotic normality of the prediction $\mu(\boldsymbol{\beta}, \boldsymbol{x}_0)$ for a test data point $\boldsymbol{x}_0$, which implies that a faithful prediction interval can be constructed for the learnt sparse neural network. Refer to Appendix A.4 for how to construct the prediction interval based on the theorem. Let $\mu_{i_1, i_2, \ldots, i_d}(\boldsymbol{\beta}, \boldsymbol{x}_0)$ denote the $d$-th order partial derivative $\frac{\partial^d \mu(\boldsymbol{\beta}, \boldsymbol{x}_0)}{\partial \boldsymbol{\beta}_{i_1} \partial \boldsymbol{\beta}_{i_2} \cdots \partial \boldsymbol{\beta}_{i_d}}$.

**Theorem 2.2.** *Assume the conditions of Theorem 2.1 and the following condition hold: $|\mu_i(\boldsymbol{\beta}^*, \boldsymbol{x}_0)| < M$, $|\mu_{i,j}(\boldsymbol{\beta}, \boldsymbol{x}_0)| < M$, $|\mu_k(\boldsymbol{\beta}, \boldsymbol{x}_0)| < M$ hold for any $i, j \in \boldsymbol{\gamma}^*, k \notin \boldsymbol{\gamma}^*$ and $\boldsymbol{\beta} \in B_{2\delta_n}(\boldsymbol{\beta}^*)$, where $M$ is as defined in Theorem 2.1. Then $\pi[\sqrt{n}(\mu(\boldsymbol{\beta}, \boldsymbol{x}_0) - \mu(\boldsymbol{\beta}^*, \boldsymbol{x}_0)) \mid D_n] \rightsquigarrow N(0, \Sigma)$, where $\Sigma = \nabla_{\boldsymbol{\gamma}^*} \mu(\boldsymbol{\beta}^*, \boldsymbol{x}_0)^T H^{-1} \nabla_{\boldsymbol{\gamma}^*} \mu(\boldsymbol{\beta}^*, \boldsymbol{x}_0)$ and $H = E(-\nabla_{\boldsymbol{\gamma}^*}^2 l_n(\boldsymbol{\beta}^*))$ is the Fisher information matrix.*

The asymptotic normality for general smooth functional has been established in [4]. For linear and quadratic functional of deep ReLU network with a spike-and-slab prior, the asymptotic normality has been established in [33]. The DNN prediction $\mu(\boldsymbol{\beta}, \boldsymbol{x}_0)$ can be viewed as a point evaluation functional over the neural network function space. However, in general, this functional is not smooth with respect to the locally asymptotic normal (LAN) norm. The results of [4] and [33] are not directly applicable for the asymptotic normality of $\mu(\boldsymbol{\beta}, \boldsymbol{x}_0)$.

## 3 Prior Annealing Algorithms for Sparse DNN Computation

As implied by Theorems 2.1 and 2.2, a consistent estimator of $(\boldsymbol{\gamma}^*, \boldsymbol{\beta}^*)$ is essential for statistical inference of the sparse DNN. Toward this goal, [31] proved that the marginal inclusion probabilities $q_i$'s can be estimated using Laplace approximation at the mode of the log-posterior. Based on this result, they proposed a multiple-run procedure. In each run, they first maximize the log-posterior by an optimization algorithm, such as SGD or Adam; then sparsify the DNN structure by truncating the weights less than a threshold to zero, where the threshold is calculated from the prior (3) based on the Laplace approximation theory; and then refine the weights of the sparsified DNN by running an optimization algorithm for a few iterations. Finally, they select a sparse DNN model from those obtained in the multiple runs according to their Bayesian evidence or BIC values. The BIC is suggested when the size of the sparse DNN is large.

Although the multiple-run procedure works well for many problems, it is hard to justify that it will lead to a consistent estimator of the true model $(\boldsymbol{\gamma}^*, \boldsymbol{\beta}^*)$. To tackle this issue, we propose two prior annealing algorithms, one from the frequentist perspective and one from the Bayesian perspective.

## 3.1 Prior Annealing: Frequentist Computation

It has been shown in [27, 13] that the loss of an over-parameterized DNN exhibits good properties:

($S^*$) For a fully connected DNN with an analytic activation function and a convex loss function at the output layer, if the number of hidden units of one layer is larger than the number of training points and the network structure from this layer on is pyramidal, then almost all local minima are globally optimal.

Motivated by this result, we propose a prior annealing algorithm, which is immune to local traps and aims to find a consistent estimate of $(\boldsymbol{\beta}^*, \boldsymbol{\gamma}^*)$ as defined in (2). The detailed procedure of the algorithm is given in Algorithm 1.

---

**Algorithm 1** Prior annealing: Frequentist

---

(i) (*Initial training*) Train a DNN satisfying condition (S*) such that a global optimal solution $\boldsymbol{\beta}_0 = \arg\max_{\boldsymbol{\beta}} l_n(\boldsymbol{\beta})$ is reached, which can be accomplished using SGD or Adam [16].

(ii) (*Prior annealing*) Initialize $\boldsymbol{\beta}$ at $\boldsymbol{\beta}_0$ and simulate from a sequence of distributions $\pi(\boldsymbol{\beta}|D_n, \tau, \eta^{(k)}, \sigma_{0,n}^{(k)}) \propto e^{nl_n(\boldsymbol{\beta})/\tau} \pi_k^{\eta^{(k)}/\tau}(\boldsymbol{\beta})$ for $k = 1, 2, \ldots, m$, where $0 < \eta^{(1)} \le \eta^{(2)} \le \cdots \le \eta^{(m)} = 1$, $\pi_k = \lambda_n N(0, \sigma_{1,n}^2) + (1 - \lambda_n)N(0, (\sigma_{0,n}^{(k)})^2)$, and $\sigma_{0,n}^{init} = \sigma_{0,n}^{(1)} \ge \sigma_{0,n}^{(2)} \ge \cdots \ge \sigma_{0,n}^{(m)} = \sigma_{0,n}^{end}$. The simulation can be done in an annealing manner using a stochastic gradient MCMC algorithm [34, 6, 22, 26]. After the stage $m$ has been reached, continue to run the simulated annealing algorithm by gradually decreasing the temperature $\tau$ to a very small value. Denote the resulting DNN by $\hat{\boldsymbol{\beta}} = (\hat{\boldsymbol{\beta}}_1, \hat{\boldsymbol{\beta}}_2, \ldots, \hat{\boldsymbol{\beta}}_{K_n})$.

(iii) (*Structure sparsification*) For each connection $i \in \{1, 2, \ldots, K_n\}$, set $\tilde{\boldsymbol{\gamma}}_i = 1$ if $|\hat{\boldsymbol{\beta}}_i| > \frac{\sqrt{2}\sigma_{0,n}\sigma_{1,n}}{\sqrt{\sigma_{1,n}^2 - \sigma_{0,n}^2}} \sqrt{\log\left(\frac{1-\lambda_n}{\lambda_n} \frac{\sigma_{1,n}}{\sigma_{0,n}}\right)}$ and 0 otherwise, where the threshold value of $|\hat{\boldsymbol{\beta}}_i|$ is obtained by solving $\pi(\boldsymbol{\gamma}_i = 1|\boldsymbol{\beta}_i) > 0.5$ based on the mixture Gaussian prior as in [31]. Denote the yielded sparse DNN structure by $\tilde{\boldsymbol{\gamma}}$.

(iv) (*Nonzero-weights refining*) Refine the nonzero weights of the sparsified DNN by maximizing $l_n(\boldsymbol{\beta})$. Denote the resulting estimate by $\tilde{\boldsymbol{\beta}}_{\tilde{\boldsymbol{\gamma}}}$, which represents the MLE of $\boldsymbol{\beta}^*$.

---

For Algorithm 1, the consistency of $(\tilde{\boldsymbol{\gamma}}, \tilde{\boldsymbol{\beta}}_{\tilde{\boldsymbol{\gamma}}})$ as an estimator of $(\boldsymbol{\gamma}^*, \boldsymbol{\beta}^*)$ can be proved based on Theorem 3.4 of [27] for global convergence of $\boldsymbol{\beta}_0$, the property of simulated annealing (by choosing an appropriate sequence of $\eta_k$ and a cooling schedule of $\tau$), Theorem 2.2 for consistency of structure selection, Theorem 2.3 of [31] for consistency of structure sparsification, and Theorem 2.1 of [29] for consistency of MLE under the scenario of dimension diverging. Then we can construct the confidence intervals for neural network predictions using $(\tilde{\boldsymbol{\gamma}}, \tilde{\boldsymbol{\beta}}_{\tilde{\boldsymbol{\gamma}}})$. The detailed procedure is given in supplementary material.

Intuitively, the initial training phase can reach the global optimum of the likelihood function. In the prior annealing phase, as we slowly add the effect of the prior, the landscape of the target distribution is gradually changed and the MCMC algorithm is likely to hit the region around the optimum of the target distribution. More explanations on the effect of the prior can be found in the supplementary material. In practice, let $t$ denote the step index, a simple implementation of the initial training and prior annealing phases of Algorithm 1 can be given as follows: (i) for $0 < t < T_1$, run initial training; (ii) for $T_1 \le t \le T_2$, fix $\sigma_{0,n}^{(t)} = \sigma_{0,n}^{init}$ and linearly increase $\eta_t$ by setting $\eta^{(t)} = \frac{t-T_1}{T_2-T_1}$; (iii) for $T_2 \le t \le T_3$, fix $\eta^{(t)} = 1$ and linearly decrease $\left(\sigma_{0,n}^{(t)}\right)^2$ by setting $\left(\sigma_{0,n}^{(t)}\right)^2 = \frac{T_3-t}{T_3-T_2}\left(\sigma_{0,n}^{init}\right)^2 + \frac{t-T_2}{T_3-T_2}\left(\sigma_{0,n}^{end}\right)^2$; (iv) for $t > T_3$, fix $\eta^{(t)} = 1$ and $\sigma_{0,n}^{(t)} = \sigma_{0,n}^{end}$ and gradually decrease the temperature $\tau$, e.g., setting $\tau_t = \frac{c}{t-T_3}$ for some constant $c$.

## 3.2 Prior Annealing: Bayesian Computation

For certain problems the size (or #nonzero elements) of $\gamma^*$ is large, calculation of the Fisher information matrix is difficult. In this case, the prediction uncertainty can be quantified via posterior simulations. The simulation can be started with a DNN satisfying condition (S*) and performed using a SGMCMC algorithm [22, 26] with an annealed prior as defined in step (ii) of Algorithm 1 (For Bayesian approach, we may fix the temperature $\tau = 1$). The over-parameterized structure and annealed prior make the simulations immune to local traps.

To justify the Bayesian estimator for the prediction mean and variance, we study the deviation of the path averaging estimator $\frac{1}{T}\sum_{t=1}^{T}\phi(\boldsymbol{\beta}^{(t)})$ and the posterior mean $\int \phi(\boldsymbol{\beta})\pi(\boldsymbol{\beta}|D_n, \eta^*, \sigma_{0,n}^*)d\boldsymbol{\beta}$ for some test function $\phi(\boldsymbol{\beta})$. For simplicity, we will focus on SGLD with prior annealing. Our analysis can be easily generalized to other SGMCMC algorithms [5].

For a test function $\phi(\cdot)$, the difference between $\phi(\boldsymbol{\beta})$ and $\int \phi(\boldsymbol{\beta})\pi(\boldsymbol{\beta}|D_n, \eta^*, \sigma_{0,n}^*)d\boldsymbol{\beta}$ can be characterized by the Poisson equation:

$$\mathcal{L}\psi(\boldsymbol{\beta}) = \phi(\boldsymbol{\beta}) - \int \phi(\boldsymbol{\beta})\pi(\boldsymbol{\beta}|D_n, \eta^*, \sigma_{0,n}^*)d\boldsymbol{\beta},$$

where $\psi(\cdot)$ is the solution of the Poisson equation and $\mathcal{L}$ is the infinitesimal generator of the Langevin diffusion. i.e. for the following Langevin diffusion $d\boldsymbol{\beta}^{(t)} = \nabla \log(\pi(\boldsymbol{\beta}|D_n, \eta^*, \sigma_{0,n}^*))dt + \sqrt{2}IdW_t$, where $I$ is identity matrix and $W_t$ is Brownian motion, we have

$$\mathcal{L}\psi(\boldsymbol{\beta}) := \langle\nabla\psi(\boldsymbol{\beta}), \nabla\log(\pi(\boldsymbol{\beta}|D_n, \eta^*, \sigma_{0,n}^*)) + \operatorname{tr}(\nabla^2\psi(\boldsymbol{\beta})).$$

Let $\mathcal{D}^k\psi$ denote the kth-order derivatives of $\psi$. To control the perturbation of $\phi(\boldsymbol{\beta})$, we need the following assumption about the function $\psi(\boldsymbol{\beta})$:

**Assumption 3.1.** *For $k \in \{0, 1, 2, 3\}$, $\mathcal{D}^k\psi$ exists and there exists a function $\mathcal{V}$, s.t. $||\mathcal{D}^k\psi|| \lesssim \mathcal{V}^{p_k}$ for some constant $p_k > 0$. In addition, $\mathcal{V}$ is smooth and the expectation of $\mathcal{V}^p$ on $\boldsymbol{\beta}^{(t)}$ is bounded for some $p \leq 2\max_k\{p_k\}$, i.e. $\sup_t \mathbb{E}(\mathcal{V}^p(\boldsymbol{\beta}^{(t)})) < \infty$, $\sum_{s\in(0,1)} \mathcal{V}^p(s\boldsymbol{\beta}_1 + (1-s)\boldsymbol{\beta}_2) \lesssim \mathcal{V}^p(\boldsymbol{\beta}_1) + \mathcal{V}^p(\boldsymbol{\beta}_2)$.*

In step $t$ of the SGLD algorithm, the drift term is replaced by $\nabla_{\boldsymbol{\beta}} \log \pi(\boldsymbol{\beta}^{(t)}|D_{m,n}^{(t)}, \eta^{(t)}, \sigma_{0,n}^{(t)})$, where $D_{m,n}^{(t)}$ is used to represent the mini-batch data used in step t. Let $\mathcal{L}_t$ be the corresponding infinitesimal generator. Let $\delta_t = \mathcal{L}_t - \mathcal{L}$. To quantify the effect of $\delta_t$, we introduce the following assumption:

**Assumption 3.2.** *$\boldsymbol{\beta}^{(t)}$ has bounded expectation and the expectation of log-prior is Lipschitz continuous with respect to $\sigma_{0,n}$, i.e. there exists some constant $M$ s.t. $\sup_t \mathbb{E}(|\boldsymbol{\beta}^{(t)}|) \leq M < \infty$. For all $t$, $|\mathbb{E}\log(\pi(\boldsymbol{\beta}^{(t)}|\lambda_n, \sigma_{0,n}^{(t_1)}, \sigma_{1,n})) - \mathbb{E}\log(\pi(\boldsymbol{\beta}^{(t)}|\lambda_n, \sigma_{0,n}^{(t_2)}, \sigma_{1,n}))| \leq M|\sigma_{0,n}^{(t_1)} - \sigma_{0,n}^{(t_2)}|$.*

Then we have the following theorem:

**Theorem 3.1.** *Suppose the model satisfy assumption 3.2, and a constant learning rate of $\epsilon$ is used. For a test function $\phi(\cdot)$, if the solution of the Poisson equation $\psi(\cdot)$ satisfy assumption 3.1, then*

$$\mathbb{E}\left(\frac{1}{T}\sum_{t=1}^{T-1}\phi(\boldsymbol{\beta}^{(t)}) - \int \phi(\boldsymbol{\beta})\pi(\boldsymbol{\beta}|D_n, \eta^*, \sigma_{0,n}^*)d\boldsymbol{\beta}\right) = O\left(\frac{1}{T\epsilon} + \frac{\sum_{t=0}^{T-1}(|\eta^{(t)} - \eta^*| + |\sigma_{0,n}^{(t)} - \sigma_{0,n}^*|)}{T} + \epsilon\right),$$
(5)

*where $\sigma_{0,n}^*$ is treated as a fixed constant.*

Theorem 3.1 shows that with prior annealing, the path averaging estimator can still be used for estimating the mean and variance of the prediction and constructing the confidence interval. The detailed procedure is given in supplementary material. For the case that a decaying learning rate is used, a similar theorem can be developed as in [5].

## 4 Numerical Experiments

This section illustrates the performance of the proposed method on synthetic and real data examples.[2] For the synthetic example, the frequentist algorithm is employed to construct prediction intervals.

---

[2]The code for running these experiments can be found in `https://github.com/sylydya/Sparse-Deep-Learning-A-New-Framework-Immuneto-Local-Traps-and-Miscalibration`

Table 1: Simulation Result: MSFE and MSPE were calculated by averaging over 10 datasets, and their standard deviations were given in the parentheses.

| Method | $|\hat{S}|$ | FSR | NSR | MSFE | MSPE |
|---|---|---|---|---|---|
| BNN_anneal | 5(0) | 0 | 0 | 2.353(0.296) | 2.428(0.297) |
| BNN_Evidence | 5(0) | 0 | 0 | 2.372(0.093) | 2.439(0.132) |
| Spinn | 10.7(3.874) | 0.462 | 0 | 4.157(0.219) | 4.488(0.350) |
| DNN | - | - | - | 1.1701e-5(1.1542e-6) | 16.9226(0.3230) |
| Dropout | - | - | - | 1.104(0.068) | 13.183(0.716) |
| BART50 | 16.5(1.222) | 0.727 | 0.1 | 11.182(0.334) | 12.097(0.366) |
| LASSO | 566.8(4.844) | 0.993 | 0.26 | 8.542(0.022) | 9.496(0.148) |
| SIS | 467.2(11.776) | 0.991 | 0.2 | 7.083(0.023) | 10.114(0.161) |

The real data example involves a large network, so both the frequentist and Bayesian algorithms are employed along with comparisons with some existing network pruning methods.

## 4.1 Synthetic Example

We consider a high-dimensional nonlinear regression problem, which shows that our method can identify the sparse network structure and relevant features as well as produce prediction intervals with correct coverage rates. The datasets were generated as in [31], where the explanatory variables $x_1, \ldots, x_{p_n}$ were simulated by independently generating $e, z_1, \ldots, z_{p_n}$ from $N(0, 1)$ and setting $x_i = \frac{e + z_i}{\sqrt{2}}$. The response variable was generated from a nonlinear regression model:

$$y = \frac{5x_2}{1 + x_1^2} + 5\sin(x_3 x_4) + 2x_5 + 0x_6 + \cdots + 0x_{2000} + \epsilon,$$

where $\epsilon \sim N(0, 1)$. Ten datasets were generated, each consisting of 10000 samples for training and 1000 samples for testing. This example was taken from [31], through which we show that the prior annealing method can achieve similar results with the multiple-run method proposed in [31].

We modeled the data by a DNN of structure 2000-10000-100-10-1 with tanh activation function. Here we intentionally made the network very wide in one hidden layer to satisfy the condition (S*). Algorithm 1 was employed to learn the model. The detailed setup for the experiments were given in the supplementary material. The variable selection performance were measured using the false selection rate $FSR = \frac{\sum_{i=1}^{10} |\hat{S}_i \backslash S|}{\sum_{i=1}^{10} |\hat{S}_i|}$ and negative selection rate $NSR = \frac{\sum_{i=1}^{10} |S \backslash \hat{S}_i|}{\sum_{i=1}^{10} |S|}$, where $S$ is the set of true variables, $\hat{S}_i$ is the set of selected variables from dataset $i$ and $|\hat{S}_i|$ is the size of $\hat{S}_i$. The predictive performance is measured by mean square prediction error (MSPE) and mean square fitting error (MSFE). We compare our method with the multiple-run method (BNN_evidence) [31] as well as other existing variable selection methods including Sparse input neural network(Spinn) [11], Bayesian adaptive regression tree (BART) [2], linear model with lasso penalty (LASSO) [32], and sure independence screening with SCAD penalty (SIS)[9]. To demonstrate the importance of selecting correct variables, we also compare our method with two dense model with the same network structure: DNN trained with dropout(Dropout) and DNN trained with no regularization(DNN). Detailed setups for these methods were given in the supplementary material as well. The results were summarized in Table 1. With a single run, our method BNN_anneal achieves similar result with the multiple-run method. The latter trained the model for 10 times and selected the best one using Bayesian evidence. While for Spinn (with LASSO penalty), even with over-parametrized structure, it performs worse than the sparse BNN model.

To quantify the uncertainty of the prediction, we conducted 100 experiments over different training sets as generated previously. We constructed 95% prediction intervals over 1000 test points. Over the 1000 test points, the average coverage rate of the prediction intervals is $94.72\%(0.61\%)$, where $(0.61\%)$ denote the standard deviation. Figure 1 shows the prediction intervals constructed for 20 of the testing points. Refer to the supplementary material for the detail of the computation.

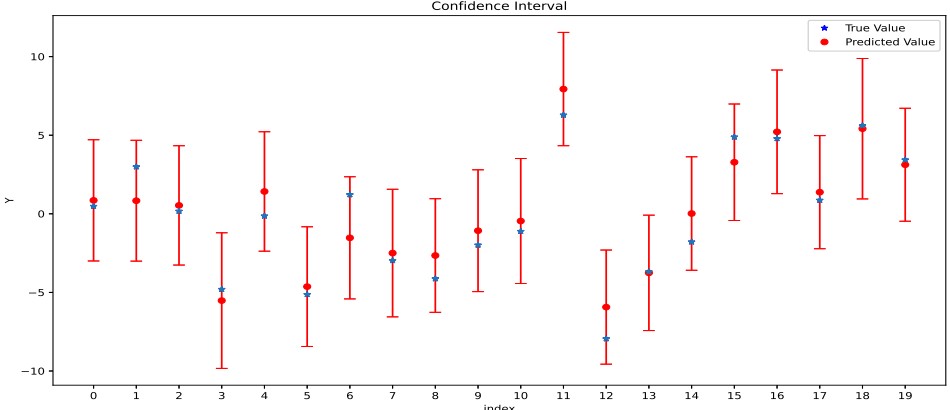

Figure 1: Prediction intervals of 20 testing points, where the y-axis is the response value, the x-axis is the index, and the blue point represents the true observation.

## 4.2 Real Data Example

As a different type of applications of the proposed method, we conducted unstructured network pruning experiments on CIFAR10 dataset[17]. Following the setup in [21], we train the residual network[15] with different networks size and pruned the network to different sparsity levels. The detailed experimental setup can be found in the supplementary material.

We compared the proposed methods, BNN_anneal (Algorithm 1) and BNN_average (averaged over last 75 networks simulated by the Bayesian version of the prior annealing algorithm), with several state-of-the-art unstructured pruning methods, including Consistent Sparse Deep Learning (BNN_BIC) [31], Dynamic pruning with feedback (DPF) [21], Dynamic Sparse Reparameterization (DSR) [25] and Sparse Momentum (SM) [7]. The results of the baseline methods were taken from [21] and [31]. The results of prediction accuracy for different models and target sparsity levels were summarized in Table 2. Due to the threshold used in step (iii) of Algorithm 1, it is hard for our method to make the pruning ratio exactly the same as the targeted one. We intentionally make the pruning ratio smaller than the target ratio, while our method still achieve better test accuracy. Compared to BNN_BIC, the test accuracy is very close, but the result of BNN_BIC is obtained by running the experiment 10 times while our method only run once. To further demonstrate that the proposed method result in better model calibration, we followed the setup of [23] and compared the proposed method with DPF on several metrics designed for model calibration, including negtive log likelihood (NLL), symmetrized, discretized KL distance between in and out of sample entropy distributions (JS-Distance), and expected calibration error (ECE). For JS-Distance, we used the test data of SVHN data set[3] as out-of-distribution samples. The results were summarized in Table 3. As discussed in [23, 14], a well calibrated model tends to have smaller NLL, larger JS-Distance and smaller ECE. The comparison shows that the proposed method outperforms DPF in most cases. In addition to the network pruning method, we also train a dense model with the standard training set up. Compared to the dense model, the sparse network has worse accuracy, but it tends to outperform the dense network in terms of ECE and JS-Distance, which indicates that sparsification is also a useful way for improving calibration of the DNN.

## 5 Conclusion

This work, together with [31], has built a solid theoretical foundation for sparse deep learning, which has successfully tamed the sparse deep neural network into the framework of statistical modeling. As implied by Lemma 2.1, Lemma 2.2, Theorem 2.1, and Theorem 2.2, the sparse DNN can be simply viewed as a nonlinear statistical model which, like a traditional statistical model, possesses many nice properties such as posterior consistency, variable selection consistency, and asymptotic

---

[3]The Street View House Numbers (SVHN) Dataset: `http://ufldl.stanford.edu/housenumbers/`

Table 2: ResNet network pruning results for CIFAR-10 data, which were calculated by averaging over 3 independent runs with the standard deviation reported in the parentheses.

| | ResNet-20 | | ResNet-32 | |
|---|---|---|---|---|
| Method | Pruning Ratio | Test Accuracy | Pruning Ratio | Test Accuracy |
| DNN_dense | 100% | 92.93(0.04) | 100% | 93.76(0.02) |
| BNN_average | 19.85%(0.18%) | 92.53(0.08) | 9.99%(0.08%) | 93.12(0.09) |
| BNN_anneal | 19.80%(0.01%) | 92.30(0.16) | 9.97%(0.03%) | 92.63(0.09) |
| BNN_BIC | 19.67%(0.05%) | 92.27(0.03) | 9.53%(0.04%) | 92.74(0.07) |
| SM | 20% | 91.54(0.16) | 10% | 91.54(0.18) |
| DSR | 20% | 91.78(0.28) | 10% | 91.41(0.23) |
| DPF | 20% | 92.17(0.21) | 10% | 92.42(0.18) |
| BNN_average | 9.88%(0.02%) | 91.65(0.08) | 4.77%(0.08%) | 91.30(0.16) |
| BNN_anneal | 9.95%(0.03%) | 91.28(0.11) | 4.88%(0.02%) | 91.17(0.08) |
| BNN_BIC | 9.55%(0.03%) | 91.27(0.05) | 4.78%(0.01%) | 91.21(0.01) |
| SM | 10% | 89.76(0.40) | 5% | 88.68(0.22) |
| DSR | 10% | 87.88(0.04) | 5% | 84.12(0.32) |
| DPF | 10% | 90.88(0.07) | 5% | 90.94(0.35) |

Table 3: ResNet network pruning results for CIFAR-10 data, which were calculated by averaging over 3 independent runs with the standard deviation reported in the parentheses.

| Method | Model | Pruning Ratio | NLL | JS-Distance | ECE |
|---|---|---|---|---|---|
| DNN_dense | ResNet20 | 100% | 0.2276(0.0021) | 7.9118(0.9316) | 0.02627(0.0005) |
| BNN_average | ResNet20 | 9.88%(0.02%) | 0.2528(0.0029) | 9.9641(0.3069) | 0.0113(0.0010) |
| BNN_anneal | ResNet20 | 9.95%(0.03%) | 0.2618(0.0037) | 10.1251(0.1797) | 0.0175(0.0011) |
| DPF | ResNet20 | 10% | 0.2833(0.0004) | 7.5712(0.4466) | 0.0294(0.0009) |
| BNN_average | ResNet20 | 19.85%(0.18%) | 0.2323(0.0033) | 7.7007(0.5374) | 0.0173(0.0014) |
| BNN_anneal | ResNet20 | 19.80%(0.01%) | 0.2441(0.0042) | 6.4435(0.2029) | 0.0233(0.0020) |
| DPF | ResNet20 | 20% | 0.2874(0.0029) | 7.7329(0.1400) | 0.0391(0.0001) |
| DNN_dense | ResNet32 | 100% | 0.2042(0.0017) | 6.7699(0.5253) | 0.02613(0.00029) |
| BNN_average | ResNet32 | 9.99%(0.08%) | 0.2116(0.0012) | 9.4549(0.5456) | 0.0132(0.0001) |
| BNN_anneal | ResNet32 | 9.97%(0.03%) | 0.2218(0.0013) | 8.5447(0.1393) | 0.0192(0.0009) |
| DPF | ResNet32 | 10% | 0.2677(0.0041) | 7.8693(0.1840) | 0.0364(0.0015) |
| BNN_average | ResNet32 | 4.77%(0.08%) | 0.2587(0.0022) | 7.0117(0.2222) | 0.0100(0.0002) |
| BNN_anneal | ResNet32 | 4.88%(0.02%) | 0.2676(0.0014) | 6.8440(0.4850) | 0.0149(0.0006) |
| DPF | ResNet32 | 5% | 0.2921(0.0067) | 6.3990(0.8384) | 0.0276(0.0019) |

normality. We have shown how the prediction uncertainty of the sparse DNN can be quantified based on the asymptotic normality theory, and provided algorithms for training sparse DNNs with theoretical guarantees for its convergence to the global optimum. The latter ensures the validity of the down-stream statistical inference.

## Acknowledgment

Funding in direct support of this work: NSF grant DMS-2015498, NIH grants R01-GM117597 and R01-GM126089, and Liang's startup fund at Purdue University.

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
