# Sparse Deep Learning: A New Framework Immune to Local Traps and Miscalibration

**Yan Sun**
Purdue University
West Lafayette, IN 47906
sun748@purdue.edu

**Wenjun Xiong**
Guangxi Normal University & Purdue University
West Lafayette, IN 47906
xiong90@purdue.edu

**Faming Liang**
Purdue University
West Lafayette, IN 47906
fmliang@purdue.edu

## 1 Proof of Theorem 2.1

*Proof.* We first define the equivalent class of neural network parameters. Given a parameter vector $\boldsymbol{\beta}$ and the corresponding structure parameter vector $\boldsymbol{\gamma}$, its equivalent class is given by

$$Q_E(\boldsymbol{\beta}, \boldsymbol{\gamma}) = \{(\tilde{\boldsymbol{\beta}}, \tilde{\boldsymbol{\gamma}}) : \nu_g(\tilde{\boldsymbol{\beta}}, \tilde{\boldsymbol{\gamma}}) = (\boldsymbol{\beta}, \boldsymbol{\gamma}), \mu(\tilde{\boldsymbol{\beta}}, \tilde{\boldsymbol{\gamma}}, \boldsymbol{x}) = \mu(\boldsymbol{\beta}, \boldsymbol{\gamma}, \boldsymbol{x}), \forall \boldsymbol{x}\},$$

where $\nu_g(\cdot)$ denotes a generic mapping that contains only the transformations of node permutation and weight sign flipping. Specifically, we define

$$Q_E^* = Q_E(\boldsymbol{\beta}^*, \boldsymbol{\gamma}^*),$$

which represents the equivalent class of *true DNN model*.

Let $B_{\delta_n}(\boldsymbol{\beta}^*) = \{\boldsymbol{\beta} : |\boldsymbol{\beta}_i - \boldsymbol{\beta}_i^*| < \delta_n, \forall i \in \boldsymbol{\gamma}^*, |\boldsymbol{\beta}_i - \boldsymbol{\beta}_i^*| < 2\sigma_{0,n}\log(\frac{\sigma_{1,n}}{\lambda_n \sigma_{0,n}}), \forall i \notin \boldsymbol{\gamma}^*\}$. By assumption C.1, $\boldsymbol{\beta}^*$ is generic (i.e. $Q_E(\boldsymbol{\beta}^*)$ contains only reparameterizations of weight sign-flipping or node permutations as defined in Feng and Simon (2017) and Fefferman (1994)) and $\min_{i \in \boldsymbol{\gamma}^*} |\boldsymbol{\beta}_i^*| - \delta_n > (C-1)\delta_n > \delta_n$, then for any $\boldsymbol{\beta}^{*(1)}, \boldsymbol{\beta}^{*(2)} \in Q_E^*$, $B_{\delta_n}(\boldsymbol{\beta}^{*(1)}) \cap B_{\delta_n}(\boldsymbol{\beta}^{*(2)}) = \emptyset$, and thus $\{\boldsymbol{\beta} : \tilde{\nu}(\boldsymbol{\beta}) \in B_{\delta_n}(\boldsymbol{\beta}^*)\} = \cup_{\boldsymbol{\beta} \in Q_E^*} B_{\delta_n}(\boldsymbol{\beta})$. In what follows, we will first show $\pi(\cup_{\boldsymbol{\beta} \in Q_E^*} B_{\delta_n}(\boldsymbol{\beta}) \mid D_n) \to 1$ as $n \to \infty$, which means the most posterior mass falls in the neighbourhood of true parameter.

Remark on the notation: $\tilde{\nu}(\cdot)$ is similar to $\nu(\cdot)$ defined in Section 2.1 of the main text. They both map the set $Q_E(\boldsymbol{\beta}, \boldsymbol{\gamma})$ to a unique network. The difference between them is that $\|\nu(\boldsymbol{\beta}) - \boldsymbol{\beta}^*\|_\infty$ may be arbitrary, but $\|\tilde{\nu}(\boldsymbol{\beta}) - \boldsymbol{\beta}^*\|_\infty$ is minimized. In other words, $\nu(\boldsymbol{\beta}, \boldsymbol{\gamma})$ is to find an arbitrary network in $Q_E(\boldsymbol{\beta}, \boldsymbol{\gamma})$ as the representative of the equivalent class, while $\tilde{\nu}(\boldsymbol{\beta}, \boldsymbol{\gamma})$ is to find a representative in $Q_E(\boldsymbol{\beta}, \boldsymbol{\gamma})$ such that the distance between $\boldsymbol{\beta}^*$ and the representative is minimized. In what follows, we will use $\tilde{\nu}(\boldsymbol{\beta})$ and $\tilde{\nu}(\boldsymbol{\gamma})$ to denote the connection weight and network structure of $\tilde{\nu}(\boldsymbol{\beta}, \boldsymbol{\gamma})$, respectively. With a slight abuse of notation, we will use $\tilde{\nu}(\boldsymbol{\beta})_i$ to denote the $i$th component of $\tilde{\nu}(\boldsymbol{\beta})$, and use $\tilde{\nu}(\boldsymbol{\gamma})_i$ to denote the $i$th component of $\tilde{\nu}(\boldsymbol{\gamma})$.

Recall that the marginal posterior inclusion probability is given by

$$q_i = \int \sum_{\boldsymbol{\gamma}} e_{i|\tilde{\nu}(\boldsymbol{\beta}, \boldsymbol{\gamma})} \pi(\boldsymbol{\gamma}|\boldsymbol{\beta}, D_n) \pi(\boldsymbol{\beta}|D_n) d\boldsymbol{\beta} = \int \pi(\tilde{\nu}(\boldsymbol{\gamma})_i = 1|\boldsymbol{\beta}) \pi(\boldsymbol{\beta}|D_n) d\boldsymbol{\beta}.$$

For the mixture Gaussian prior,

$$\pi(\boldsymbol{\gamma}_i = 1|\boldsymbol{\beta}) = \frac{1}{1 + \frac{\sigma_{1,n}(1-\lambda_n)}{\sigma_{0,n}\lambda_n} e^{-(\frac{1}{2\sigma_{0,n}^2} - \frac{1}{2\sigma_{1,n}^2})\boldsymbol{\beta}_i^2}},$$

35th Conference on Neural Information Processing Systems (NeurIPS 2021).

which increases with respect to $|\boldsymbol{\beta}_i|$. In particular, if $|\boldsymbol{\beta}_i| > 2\sigma_{0,n}\log(\frac{\sigma_{1,n}}{\lambda_n\sigma_{0,n}})$, then $\pi(\boldsymbol{\gamma}_i = 1|\boldsymbol{\beta}) > \frac{1}{2}$.

For the mixture Gaussian prior,

$$\pi(\boldsymbol{\beta} \notin \cup_{\boldsymbol{\beta}\in Q_E^*}B_{\delta_n}(\boldsymbol{\beta}) \mid D_n)$$
$$\leq \pi(\exists i \notin \boldsymbol{\gamma}^*, |\tilde{\nu}(\boldsymbol{\beta})_i| > 2\sigma_{0,n}\log(\frac{\sigma_{1,n}}{\lambda_n\sigma_{0,n}}) \mid D_n) + \pi(\exists i \in \boldsymbol{\gamma}^*, |\tilde{\nu}(\boldsymbol{\beta})_i - \boldsymbol{\beta}_i^*| > \delta_n \mid D_n).$$

For the first term, note that for a given $i \notin \boldsymbol{\gamma}^*$,

$$\pi(|\tilde{\nu}(\boldsymbol{\beta})_i| > 2\sigma_{0,n}\log(\frac{\sigma_{1,n}}{\lambda_n\sigma_{0,n}}) \mid D_n) \leq \pi(\pi(\tilde{\nu}(\boldsymbol{\gamma})_i = 1|\boldsymbol{\beta}) > \frac{1}{2} \mid D_n)$$

$$\leq 2\int \pi(\tilde{\nu}(\boldsymbol{\gamma})_i = 1|\boldsymbol{\beta})\pi(\boldsymbol{\beta}|D_n)d\boldsymbol{\beta}$$

$$\leq 2\rho(\epsilon_n) + 2\pi(d(p_{\boldsymbol{\beta}}, p_{\mu^*}) \geq \epsilon_n \mid D_n) \to 0.$$

Then we have

$$\pi(\exists i \notin \boldsymbol{\gamma}^*, |\tilde{\nu}(\boldsymbol{\beta})_i| > 2\sigma_{0,n}\log(\frac{\sigma_{1,n}}{\lambda_n\sigma_{0,n}}) \mid D_n) = \pi(\max_{i\notin\boldsymbol{\gamma}^*}|\tilde{\nu}(\boldsymbol{\beta})_i| > 2\sigma_{0,n}\log(\frac{\sigma_{1,n}}{\lambda_n\sigma_{0,n}}) \mid D_n)$$

$$\leq \pi(\max_{i\notin\boldsymbol{\gamma}^*}\pi(\tilde{\nu}(\boldsymbol{\gamma})_i = 1|\boldsymbol{\beta}) > \frac{1}{2} \mid D_n)$$

$$\leq \sum_{i\notin\boldsymbol{\gamma}^*}\pi(\pi(\tilde{\nu}(\boldsymbol{\gamma})_i = 1|\boldsymbol{\beta}) > \frac{1}{2} \mid D_n)$$

$$\leq 2K_n\rho(\epsilon_n) + 2K_n\pi(d(p_{\boldsymbol{\beta}}, p_{\mu^*}) \geq \epsilon_n \mid D_n) \to 0.$$

For the second term, by condition C.1 and Lemma 2.1,

$$\pi(\exists i \in \boldsymbol{\gamma}^*, |\tilde{\nu}(\boldsymbol{\beta})_i - \boldsymbol{\beta}_i^*| > \delta_n \mid D_n) = \pi(\max_{i\in\boldsymbol{\gamma}^*}|\tilde{\nu}(\boldsymbol{\beta})_i - \boldsymbol{\beta}_i^*| > \delta_n \mid D_n)$$

$$= \pi(\max_{i\in\boldsymbol{\gamma}^*}|\tilde{\nu}(\boldsymbol{\beta})_i - \boldsymbol{\beta}_i^*| > \delta_n, d(p_{\boldsymbol{\beta}}, p_{\mu^*}) \leq \epsilon_n \mid D_n)$$

$$+ \pi(\max_{i\in\boldsymbol{\gamma}^*}|\tilde{\nu}(\boldsymbol{\beta})_i - \boldsymbol{\beta}_i^*| > \delta_n, d(p_{\boldsymbol{\beta}}, p_{\mu^*}) \geq \epsilon_n \mid D_n)$$

$$\leq \pi(A(\epsilon_n, \delta_n) \mid D_n) + \pi(d(p_{\boldsymbol{\beta}}, p_{\mu^*}) \geq \epsilon_n \mid D_n) \to 0.$$

Summarizing the above two terms, we have $\pi(\cup_{\boldsymbol{\beta}\in Q_E^*}B_{\delta_n}(\boldsymbol{\beta}) \mid D_n) \to 1$.

Let $Q_n = |Q_E^*|$ be the number of equivalent *true DNN model*. By the generic assumption of $\boldsymbol{\beta}^*$, for any $\boldsymbol{\beta}^{*(1)}, \boldsymbol{\beta}^{*(2)} \in Q_E^*$, $B_{\delta_n}(\boldsymbol{\beta}^{*(1)}) \cap B_{\delta_n}(\boldsymbol{\beta}^{*(2)}) = \emptyset$. Then in $B_{\delta_n}(\boldsymbol{\beta}^*)$, the posterior density of $\tilde{\nu}(\boldsymbol{\beta})$ is $Q_n$ times the posterior density of $\boldsymbol{\beta}$. Then for a function $f(\cdot)$ of $\tilde{\nu}(\boldsymbol{\beta})$, by changing variable,

$$\int_{\tilde{\nu}(\boldsymbol{\beta})\in B_{\delta_n}(\boldsymbol{\beta}^*)} f(\tilde{\nu}(\boldsymbol{\beta}))\pi(\tilde{\nu}(\boldsymbol{\beta})|D_n)d\tilde{\nu}(\boldsymbol{\beta}) = Q_n\int_{B_{\delta_n}(\boldsymbol{\beta}^*)} f(\boldsymbol{\beta})\pi(\boldsymbol{\beta}|D_n)d\boldsymbol{\beta}.$$

Thus, we only need to consider the integration on $B_{\delta_n}(\boldsymbol{\beta}^*)$. Define

$$\hat{\boldsymbol{\beta}}_i = \begin{cases} \boldsymbol{\beta}_i^* - \sum_{j\in\boldsymbol{\gamma}^*} h^{i,j}(\boldsymbol{\beta}^*)h_j(\boldsymbol{\beta}^*), & \forall i \in \boldsymbol{\gamma}^*, \\ 0, & \forall i \notin \boldsymbol{\gamma}^*. \end{cases}$$

We will first prove that for any real vector $\boldsymbol{t}$,

$$E(e^{\sqrt{n}\boldsymbol{t}^T(\tilde{\nu}(\boldsymbol{\beta})-\hat{\boldsymbol{\beta}})} \mid D_n, B_{\delta_n}(\boldsymbol{\beta}^*)) := \frac{\int_{B_{\delta_n}(\boldsymbol{\beta}^*)} e^{\sqrt{n}\boldsymbol{t}^T(\tilde{\nu}(\boldsymbol{\beta})-\hat{\boldsymbol{\beta}})}\pi(\tilde{\nu}(\boldsymbol{\beta})|D_n)d\tilde{\nu}(\boldsymbol{\beta})}{\int_{B_{\delta_n}(\boldsymbol{\beta}^*)} \pi(\tilde{\nu}(\boldsymbol{\beta})|D_n)d\tilde{\nu}(\boldsymbol{\beta})}$$

$$= \frac{\int_{B_{\delta_n}(\boldsymbol{\beta}^*)} e^{\sqrt{n}\boldsymbol{t}^T(\boldsymbol{\beta}-\hat{\boldsymbol{\beta}})}e^{nl_n(\boldsymbol{\beta})}\pi(\boldsymbol{\beta})d\boldsymbol{\beta}}{\int_{B_{\delta_n}(\boldsymbol{\beta}^*)} e^{nl_n(\boldsymbol{\beta})}\pi(\boldsymbol{\beta})d\boldsymbol{\beta}} \tag{1}$$

$$= e^{\frac{1}{2}\boldsymbol{t}^T\boldsymbol{V}\boldsymbol{t}+o_{P^*}(1)}.$$

For any $\boldsymbol{\beta} \in B_{\delta_n}(\boldsymbol{\beta}^*)$, we have

$$|\sqrt{n}(\boldsymbol{t}^T(\boldsymbol{\beta} - \boldsymbol{\beta}_{\boldsymbol{\gamma}^*}))| \leq \sqrt{n}K_n\|\boldsymbol{t}\|_\infty 2\sigma_{0,n} \log(\frac{\sigma_{1,n}}{\lambda_n\sigma_{0,n}}) = o(1),$$

$$|n(l_n(\boldsymbol{\beta}) - l_n(\boldsymbol{\beta}_{\boldsymbol{\gamma}^*}))| = |n\sum_{i\notin\boldsymbol{\gamma}^*}\boldsymbol{\beta}_i(h_i(\tilde{\boldsymbol{\beta}}))| \leq nK_nM2\sigma_{0,n}\log(\frac{\sigma_{1,n}}{\lambda_n\sigma_{0,n}}) = o(1).$$

Then, we have

$$\sqrt{n}\boldsymbol{t}^T(\boldsymbol{\beta} - \hat{\boldsymbol{\beta}}) = \sqrt{n}\boldsymbol{t}^T(\boldsymbol{\beta} - \boldsymbol{\beta}_{\boldsymbol{\gamma}^*} + \boldsymbol{\beta}_{\boldsymbol{\gamma}^*} - \boldsymbol{\beta}^*) + \sqrt{n}\sum_{i,j\in\boldsymbol{\gamma}^*}h^{i,j}(\boldsymbol{\beta}^*)\boldsymbol{t}_jh_i(\boldsymbol{\beta}^*))$$

$$= o(1) + \sqrt{n}\sum_{i\in\boldsymbol{\gamma}^*}(\boldsymbol{\beta}_i - \boldsymbol{\beta}_i^*)\boldsymbol{t}_i + \sqrt{n}\sum_{i,j\in\boldsymbol{\gamma}^*}h^{i,j}(\boldsymbol{\beta}^*)\boldsymbol{t}_jh_i(\boldsymbol{\beta}^*), \tag{2}$$

$$nl_n(\boldsymbol{\beta}) - nl_n(\boldsymbol{\beta}^*) = n(l_n(\boldsymbol{\beta}) - l_n(\boldsymbol{\beta}_{\boldsymbol{\gamma}^*}) + l_n(\boldsymbol{\beta}_{\boldsymbol{\gamma}^*}) - nl_n(\boldsymbol{\beta}^*))$$

$$= o(1) + n\sum_{i\in\boldsymbol{\gamma}^*}(\boldsymbol{\beta}_i - \boldsymbol{\beta}_i^*)h_i(\boldsymbol{\beta}^*) + \frac{n}{2}\sum_{i,j\in\boldsymbol{\gamma}^*}h_{i,j}(\boldsymbol{\beta}^*)(\boldsymbol{\beta}_i - \boldsymbol{\beta}_i^*)(\boldsymbol{\beta}_j - \boldsymbol{\beta}_j^*)$$

$$+ \frac{n}{6}\sum_{i,j,k\in\boldsymbol{\gamma}^*}h_{i,j,k}(\check{\boldsymbol{\beta}})(\boldsymbol{\beta}_i - \boldsymbol{\beta}_i^*)(\boldsymbol{\beta}_j - \boldsymbol{\beta}_j^*)(\boldsymbol{\beta}_k - \boldsymbol{\beta}_k^*), \tag{3}$$

where $\check{\boldsymbol{\beta}}$ is a point between $\boldsymbol{\beta}_{\boldsymbol{\gamma}^*}$ and $\boldsymbol{\beta}^*$. Note that for $\boldsymbol{\beta} \in B_{\delta_n}(\boldsymbol{\beta}^*)$, $|\boldsymbol{\beta}_i - \boldsymbol{\beta}_i^*| \leq \delta_n \lesssim \frac{1}{\sqrt[3]{n}r_n}$, we have $\frac{n}{6}\sum_{i,j,k\in\boldsymbol{\gamma}^*}h_{i,j,k}(\check{\boldsymbol{\beta}})(\boldsymbol{\beta}_i - \boldsymbol{\beta}_i^*)(\boldsymbol{\beta}_j - \boldsymbol{\beta}_j^*)(\boldsymbol{\beta}_k - \boldsymbol{\beta}_k^*) = o(1)$.

Let $\boldsymbol{\beta}^{(t)}$ be network parameters satisfying $\boldsymbol{\beta}_i^{(t)} = \boldsymbol{\beta}_i + \frac{1}{\sqrt{n}}\sum_{j\in\boldsymbol{\gamma}^*}h^{i,j}(\boldsymbol{\beta}^*)\boldsymbol{t}_j, \forall i \in \boldsymbol{\gamma}^*$ and $\boldsymbol{\beta}_i^{(t)} = \boldsymbol{\beta}_i, \forall i \notin \boldsymbol{\gamma}^*$. Note that $\frac{1}{\sqrt{n}}\sum_{j\in\boldsymbol{\gamma}^*}h^{i,j}(\boldsymbol{\beta}^*)\boldsymbol{t}_j \leq \frac{r_n\|\boldsymbol{t}\|_\infty M}{\sqrt{n}} \lesssim \delta_n$, for large enough $n$, $|\boldsymbol{\beta}_i^{(t)}| < 2\delta_n$ $\forall i \in \boldsymbol{\gamma}^*$. Thus, we have

$$nl_n(\boldsymbol{\beta}^{(t)}) - nl_n(\boldsymbol{\beta}^*) = n(l_n(\boldsymbol{\beta}^{(t)}) - l_n(\boldsymbol{\beta}_{\boldsymbol{\gamma}^*}^{(t)}) + l_n(\boldsymbol{\beta}_{\boldsymbol{\gamma}^*}^{(t)}) - nl_n(\boldsymbol{\beta}^*))$$

$$= o(1) + n\sum_{i\in\boldsymbol{\gamma}^*}(\boldsymbol{\beta}_i^{(t)} - \boldsymbol{\beta}_i^*)h_i(\boldsymbol{\beta}^*) + \frac{n}{2}\sum_{i,j\in\boldsymbol{\gamma}^*}h_{i,j}(\boldsymbol{\beta}^*)(\boldsymbol{\beta}_i^{(t)} - \boldsymbol{\beta}_i^*)(\boldsymbol{\beta}_j^{(t)} - \boldsymbol{\beta}_j^*)$$

$$= o(1) + n\sum_{i\in\boldsymbol{\gamma}^*}(\boldsymbol{\beta}_i - \boldsymbol{\beta}_i^*)h_i(\boldsymbol{\beta}^*) + \frac{n}{2}\sum_{i,j\in\boldsymbol{\gamma}^*}h_{i,j}(\boldsymbol{\beta}^*)(\boldsymbol{\beta}_i - \boldsymbol{\beta}_i^*)(\boldsymbol{\beta}_j - \boldsymbol{\beta}_j^*)$$

$$+ \sqrt{n}\sum_{i,j\in\boldsymbol{\gamma}^*}h^{i,j}(\boldsymbol{\beta}^*)\boldsymbol{t}_jh_i(\boldsymbol{\beta}^*) + \sqrt{n}\sum_{i\in\boldsymbol{\gamma}^*}(\boldsymbol{\beta}_i - \boldsymbol{\beta}_i^*)\boldsymbol{t}_i + \frac{1}{2}\sum_{i,j\in\boldsymbol{\gamma}^*}h^{i,j}(\boldsymbol{\beta}^*)\boldsymbol{t}_i\boldsymbol{t}_j$$

$$= o(1) + \sqrt{n}\boldsymbol{t}^T(\boldsymbol{\beta} - \hat{\boldsymbol{\beta}}) + nl_n(\boldsymbol{\beta}) - nl_n(\boldsymbol{\beta}^*) + \frac{1}{2}\sum_{i,j\in\boldsymbol{\gamma}^*}h^{i,j}(\boldsymbol{\beta}^*)\boldsymbol{t}_i\boldsymbol{t}_j, \tag{4}$$

where the last equality is derived by replacing appropriate terms by $\sqrt{n}\boldsymbol{t}^T(\boldsymbol{\beta} - \hat{\boldsymbol{\beta}})$ and $nl_n(\boldsymbol{\beta}) - nl_n(\boldsymbol{\beta}^*)$ based on (2) and (3), respectively; and the third equality is derived based on the following calculation:

$$\frac{n}{2}\sum_{i,j\in\boldsymbol{\gamma}^*}h_{i,j}(\boldsymbol{\beta}^*)(\boldsymbol{\beta}_i^{(t)} - \boldsymbol{\beta}_i^*)(\boldsymbol{\beta}_j^{(t)} - \boldsymbol{\beta}_j^*)$$

$$= \frac{n}{2}\sum_{i,j\in\boldsymbol{\gamma}^*}h_{i,j}(\boldsymbol{\beta}^*)(\boldsymbol{\beta}_i - \boldsymbol{\beta}_i^* + \frac{1}{\sqrt{n}}\sum_{k\in\boldsymbol{\gamma}^*}h^{i,k}(\boldsymbol{\beta}^*)\boldsymbol{t}_k)(\boldsymbol{\beta}_j - \boldsymbol{\beta}_j^* + \frac{1}{\sqrt{n}}\sum_{k\in\boldsymbol{\gamma}^*}h^{j,k}(\boldsymbol{\beta}^*)\boldsymbol{t}_k)$$

$$= \frac{n}{2}\sum_{i,j\in\boldsymbol{\gamma}^*}h_{i,j}(\boldsymbol{\beta}^*)(\boldsymbol{\beta}_i - \boldsymbol{\beta}_i^*)(\boldsymbol{\beta}_j - \boldsymbol{\beta}_j^*) + 2 \times \frac{n}{2}\sum_{i,j\in\boldsymbol{\gamma}^*}h_{i,j}(\boldsymbol{\beta}^*)\frac{1}{\sqrt{n}}\sum_{k\in\boldsymbol{\gamma}^*}h^{i,k}(\boldsymbol{\beta}^*)\boldsymbol{t}_k(\boldsymbol{\beta}_j - \boldsymbol{\beta}_j^*)$$

$$+ \frac{n}{2}\sum_{i,j\in\boldsymbol{\gamma}^*}h_{i,j}(\boldsymbol{\beta}^*)(\frac{1}{\sqrt{n}}\sum_{k\in\boldsymbol{\gamma}^*}h^{i,k}(\boldsymbol{\beta}^*)\boldsymbol{t}_k)(\frac{1}{\sqrt{n}}\sum_{k\in\boldsymbol{\gamma}^*}h^{j,k}(\boldsymbol{\beta}^*)\boldsymbol{t}_k)$$

$$= \frac{n}{2}\sum_{i,j\in\boldsymbol{\gamma}^*}h_{i,j}(\boldsymbol{\beta}^*)(\boldsymbol{\beta}_i - \boldsymbol{\beta}_i^*)(\boldsymbol{\beta}_j - \boldsymbol{\beta}_j^*) + \sqrt{n}\sum_{i\in\boldsymbol{\gamma}^*}(\boldsymbol{\beta}_i - \boldsymbol{\beta}_i^*)\boldsymbol{t}_i + \frac{1}{2}\sum_{i,j\in\boldsymbol{\gamma}^*}h^{i,j}(\boldsymbol{\beta}^*)\boldsymbol{t}_i\boldsymbol{t}_j, \tag{5}$$

where the second and third terms in the last equality are derived based on the relation $\sum_{i \in \gamma^*} h_{i,j}(\boldsymbol{\beta}^*) h^{i,k}(\boldsymbol{\beta}^*) = \delta_{j,k}$, where $\delta_{j,k} = 1$ if $j = k$, $\delta_{j,k} = 0$ if $j \neq k$.

By rearranging the terms in (4), we have

$$\int_{B_{\delta_n}(\boldsymbol{\beta}^*)} \exp\{\sqrt{n}\boldsymbol{t}^T(\boldsymbol{\beta} - \hat{\boldsymbol{\beta}}) + nl_n(\boldsymbol{\beta})\}\pi(\boldsymbol{\beta})d\boldsymbol{\beta}$$

$$= \exp\left\{-\frac{1}{2}\sum_{i,j \in \gamma^*} h^{i,j}(\boldsymbol{\beta}^*)\boldsymbol{t}_i\boldsymbol{t}_j + o(1)\right\} \int_{B_{\delta_n}(\boldsymbol{\beta}^*)} e^{nl_n(\boldsymbol{\beta}^{(t)})}\pi(\boldsymbol{\beta})d\boldsymbol{\beta}.$$

For $\boldsymbol{\beta} \in B_{\delta_n}(\boldsymbol{\beta}^*), i \in \gamma^*$, by Assumption C.1, there exists a constant $C > 2$ such that

$$|\boldsymbol{\beta}_i^{(t)}| \geq |\boldsymbol{\beta}_i| - \frac{r_n\|\boldsymbol{t}\|_\infty M}{\sqrt{n}} \geq |\boldsymbol{\beta}_i^*| - 2\delta_n \geq (C-2)\delta_n \gtrsim \frac{r_n}{\sqrt{n}}$$

$$\gtrsim \sqrt{\left(\frac{1}{2\sigma_{0,n}^2} - \frac{1}{2\sigma_{1,n}^2}\right)^{-1} \log\left(\frac{r_n(1-\lambda_n)\sigma_{1,n}}{\sigma_{0,n}\lambda_n}\right)}.$$

Then we have

$$\frac{\sigma_{1,n}(1-\lambda_n)}{\sigma_{0,n}\lambda_n}e^{-(\frac{1}{2\sigma_{0,n}^2} - \frac{1}{2\sigma_{1,n}^2})(\boldsymbol{\beta}_i^{(t)})^2} \lesssim \frac{1}{r_n}.$$

It is easy to see that the above formula also holds if we replace $\boldsymbol{\beta}_i^{(t)}$ by $\boldsymbol{\beta}_i$. Note that the mixture Gaussian prior of $\boldsymbol{\beta}_i$ can be written as

$$\pi(\boldsymbol{\beta}_i) = \frac{\lambda_n}{\sqrt{2\pi}\sigma_{1,n}}e^{-\frac{\boldsymbol{\beta}_i^2}{2\sigma_{1,n}^2}}\left(1 + \frac{\sigma_{1,n}(1-\lambda_n)}{\sigma_{0,n}\lambda_n}e^{-(\frac{1}{2\sigma_{0,n}^2} - \frac{1}{2\sigma_{1,n}^2})\boldsymbol{\beta}_i^2}\right).$$

Since $|\boldsymbol{\beta}_i - \boldsymbol{\beta}_i^{(t)}| \lesssim \delta_n \lesssim \frac{1}{\sqrt[3]{nr_n}}$, $|\boldsymbol{\beta}_i + \boldsymbol{\beta}_i^{(t)}| < 2E_n + 3\delta_n \lesssim E_n$, and $\frac{1}{\sigma_{1,n}^2} \lesssim \frac{H_n \log(n) + \log(\bar{L})}{E_n^2}$, we have

$$\frac{r_n}{\sigma_{1,n}^2}(\boldsymbol{\beta}_i - \boldsymbol{\beta}_i^{(t)})(\boldsymbol{\beta}_i + \boldsymbol{\beta}_i^{(t)}) = \frac{H_n \log(n) + \log(\bar{L})}{n^{C_1+1/3}} = o(1),$$

by the condition $C_1 > 2/3$ and $H_n \log(n) + \log(\bar{L}) \prec n^{1-\epsilon}$. Thus, $\frac{\pi(\boldsymbol{\beta})}{\pi(\boldsymbol{\beta}^{(t)})} = \prod_{i \in \gamma^*} \frac{\pi(\boldsymbol{\beta}_i)}{\pi(\boldsymbol{\beta}_i^{(t)})} = 1 + o(1)$, and

$$\int_{B_{\delta_n}(\boldsymbol{\beta}^*)} e^{nl_n(\boldsymbol{\beta}^{(t)})}\pi(\boldsymbol{\beta})d\boldsymbol{\beta} = (1 + o(1)) \int_{\boldsymbol{\beta}^{(t)} \in B_{\delta_n}(\boldsymbol{\beta}^*)} e^{nl_n(\boldsymbol{\beta}^{(t)})}\pi(\boldsymbol{\beta}^{(t)})d\boldsymbol{\beta}^{(t)} \tag{6}$$

$$= (1 + o(1))C_N\pi(\boldsymbol{\beta}^{(t)} \in B_{\delta_n}(\boldsymbol{\beta}^*) \mid D_n),$$

where $C_N$ is the normalizing constant of the posterior. Note that $\|\boldsymbol{\beta}^{(t)} - \boldsymbol{\beta}\|_\infty \lesssim \delta_n$, we have $\pi(\boldsymbol{\beta}^{(t)} \in B_{\delta_n}(\boldsymbol{\beta}^*) \mid D_n) \to \pi(\boldsymbol{\beta} \in B_{\delta_n}(\boldsymbol{\beta}^*) \mid D_n)$. Moreover, since $-\frac{1}{2}\sum_{i,j \in \gamma^*} h^{i,j}(\boldsymbol{\beta}^*)\boldsymbol{t}_i\boldsymbol{t}_j \to \frac{1}{2}\boldsymbol{t}^T\boldsymbol{V}\boldsymbol{t}$, we have

$$E(e^{\sqrt{n}\boldsymbol{t}^T(\tilde{\nu}(\boldsymbol{\beta}) - \hat{\boldsymbol{\beta}})} \mid D_n, B_{\delta_n}(\boldsymbol{\beta}^*)) = \frac{\int_{B_{\delta_n}(\boldsymbol{\beta}^*)} e^{\sqrt{n}\boldsymbol{t}^T(\boldsymbol{\beta} - \hat{\boldsymbol{\beta}})}e^{nh_n(\boldsymbol{\beta})}\pi(\boldsymbol{\beta})d\boldsymbol{\beta}}{\int_{B_{\delta_n}(\boldsymbol{\beta}^*)} e^{nh_n(\boldsymbol{\beta})}\pi(\boldsymbol{\beta})d\boldsymbol{\beta}} = e^{\frac{\boldsymbol{t}^T\boldsymbol{V}\boldsymbol{t}}{2} + o_{P^*}(1)}.$$

Combining the above result with the fact that $\pi(\tilde{\nu}(\boldsymbol{\beta}) \in B_{\delta_n}(\boldsymbol{\beta}^*) \mid D_n) \to 1$, by section 1 of Castillo and Rousseau (2015), we have

$$\pi[\sqrt{n}(\tilde{\nu}(\boldsymbol{\beta}) - \hat{\boldsymbol{\beta}}) \mid D_n] \rightsquigarrow N(0, \boldsymbol{V}).$$

We will then show that $\hat{\boldsymbol{\beta}}$ will converge to $\boldsymbol{\beta}^*$, then essentially we can replace $\hat{\boldsymbol{\beta}}$ by $\boldsymbol{\beta}^*$ in the above result. Let $\boldsymbol{\Theta}_{\gamma^*} = \{\boldsymbol{\beta} : \boldsymbol{\beta}_i = 0, \forall i \notin \gamma^*\}$ be the parameter space given the model $\gamma^*$, and let $\hat{\boldsymbol{\beta}}_{\gamma^*}$ be the maximum likelihood estimator given the model $\gamma^*$, i.e.

$$\hat{\boldsymbol{\beta}}_{\gamma^*} = \arg\max_{\boldsymbol{\beta} \in \boldsymbol{\Theta}_{\gamma^*}} l_n(\boldsymbol{\beta}).$$

Given condition C.3 and by Theorem 2.1 of Portnoy (1988), we have $||\hat{\boldsymbol{\beta}}_{\boldsymbol{\gamma}^*} - \boldsymbol{\beta}^*|| = O(\sqrt{\frac{r_n}{n}}) = o(1)$. Note that $h_i(\hat{\boldsymbol{\beta}}_{\boldsymbol{\gamma}^*}) = 0$ as $\hat{\boldsymbol{\beta}}_{\boldsymbol{\gamma}^*}$ is maximum likelihood estimator. Then for any $i \in \boldsymbol{\gamma}^*$, $|h_i(\boldsymbol{\beta}^*)| = |h_i(\hat{\boldsymbol{\beta}}_{\boldsymbol{\gamma}^*}) - h_i(\boldsymbol{\beta}^*)| = |\sum_{j \in \boldsymbol{\gamma}^*} h_{ij}(\tilde{\boldsymbol{\beta}})((\hat{\boldsymbol{\beta}}_{\boldsymbol{\gamma}^*})_j - \boldsymbol{\beta}_j^*)| \le M||\hat{\boldsymbol{\beta}}_{\boldsymbol{\gamma}^*} - \boldsymbol{\beta}^*||_1 = O(\sqrt{\frac{r_n}{n}})$.

Then for any $i, j \in \boldsymbol{\gamma}^*$, we have $\sum_{j \in \boldsymbol{\gamma}^*} h^{i,j}(\boldsymbol{\beta}^*) h_j(\boldsymbol{\beta}^*) = O(\sqrt{\frac{r_n^3}{n}}) = o(1)$. By the definition of $\hat{\boldsymbol{\beta}}$, we have $\hat{\boldsymbol{\beta}} - \boldsymbol{\beta}^* = o(1)$. Therefore, we have

$$\pi[\sqrt{n}(\tilde{\nu}(\boldsymbol{\beta}) - \boldsymbol{\beta}^*) \mid D_n] \rightsquigarrow N(0, \boldsymbol{V}).$$

## 2 Proof of Theorem 2.2

*Proof.* The proof of Theorem 2.2 can be done using the same strategy as that used in proving Theorem 2.1. Here we provide a simpler proof using the result of Theorem 2.1. The notations we used in this proof are the same as in the proof of Theorem 2.1. In the proof of Theorem 2.1, we have shown that $\pi(\tilde{\nu}(\boldsymbol{\beta}) \in B_{\delta_n}(\boldsymbol{\beta}^*) \mid D_n) \to 1$. Note that $\mu(\boldsymbol{\beta}, \boldsymbol{x}_0) = \mu(\tilde{\nu}(\boldsymbol{\beta}), \boldsymbol{x}_0)$. We only need to consider $\boldsymbol{\beta} \in B_{\delta_n}(\boldsymbol{\beta}^*)$. For $\boldsymbol{\beta} \in B_{\delta_n}(\boldsymbol{\beta}^*)$, we have

$$\begin{aligned}
&\sqrt{n}(\mu(\boldsymbol{\beta}, \boldsymbol{x}_0) - \mu(\boldsymbol{\beta}^*, \boldsymbol{x}_0)) \\
=&\sqrt{n}(\mu(\boldsymbol{\beta}, \boldsymbol{x}_0) - \mu(\boldsymbol{\beta}_{\boldsymbol{\gamma}^*}, \boldsymbol{x}_0) + \mu(\tilde{\nu}(\boldsymbol{\beta}_{\boldsymbol{\gamma}^*}), \boldsymbol{x}_0) - \mu(\boldsymbol{\beta}^*, \boldsymbol{x}_0)).
\end{aligned}$$

Since $\boldsymbol{\beta} \in B_{\delta_n}(\boldsymbol{\beta}^*)$, for $i \notin \boldsymbol{\gamma}^*$, $|\boldsymbol{\beta}_i| < 2\sigma_{0,n} \log(\frac{\sigma_{1,n}}{\lambda_n \sigma_{0,n}})$; and for $i \in \boldsymbol{\gamma}^*$, $|\tilde{\nu}(\boldsymbol{\beta})_i - \boldsymbol{\beta}_i^*| < \delta \lesssim \frac{1}{\sqrt[3]{n r_n}}$. Therefore,

$$|\sqrt{n}\mu(\boldsymbol{\beta}, \boldsymbol{x}_0) - \mu(\boldsymbol{\beta}_{\boldsymbol{\gamma}^*}, \boldsymbol{x}_0))| = |\sqrt{n} \sum_{i \notin \boldsymbol{\gamma}^*} \boldsymbol{\beta}_i(\mu_i(\tilde{\boldsymbol{\beta}}, \boldsymbol{x}_0))| \le \sqrt{n} K_n M 2\sigma_{0,n} \log(\frac{\sigma_{1,n}}{\lambda_n \sigma_{0,n}}) = o(1),$$

where $\mu_i(\boldsymbol{\beta}, \boldsymbol{x}_0)$ denotes the first derivative of $\mu(\boldsymbol{\beta}, \boldsymbol{x}_0)$ with respect to the $i$th component of $\boldsymbol{\beta}$, and $\tilde{\boldsymbol{\beta}}$ denotes a point between $\boldsymbol{\beta}$ and $\boldsymbol{\beta}_{\boldsymbol{\gamma}^*}$. Further,

$$\begin{aligned}
&\mu(\tilde{\nu}(\boldsymbol{\beta}_{\boldsymbol{\gamma}^*}), \boldsymbol{x}_0) - \mu(\boldsymbol{\beta}^*, \boldsymbol{x}_0) \\
=&\sqrt{n} \sum_{i \in \boldsymbol{\gamma}^*} (\tilde{\nu}(\boldsymbol{\beta})_i - \boldsymbol{\beta}_i^*)\mu_i(\boldsymbol{\beta}^*, \boldsymbol{x}_0) + \sqrt{n} \sum_{i \in \boldsymbol{\gamma}^*} \sum_{j \in \boldsymbol{\gamma}^*} (\tilde{\nu}(\boldsymbol{\beta})_i - \boldsymbol{\beta}_i^*)\mu_{i,j}(\check{\boldsymbol{\beta}}, \boldsymbol{x}_0)(\tilde{\nu}(\boldsymbol{\beta})_j - \boldsymbol{\beta}_j^*) \\
=&\sqrt{n} \sum_{i \in \boldsymbol{\gamma}^*} ((\tilde{\nu}(\boldsymbol{\beta})_i - \boldsymbol{\beta}_i^*)\mu_i(\boldsymbol{\beta}^*, \boldsymbol{x}_0) + o(1),
\end{aligned}$$

where $\mu_{i,j}(\boldsymbol{\beta}, \boldsymbol{x}_0)$ denotes the second derivative of $\mu(\boldsymbol{\beta}, \boldsymbol{x}_0)$ with respect to the $i$th and $j$th components of $\boldsymbol{\beta}$ and $\check{\boldsymbol{\beta}}$ is a point between $\tilde{\nu}(\boldsymbol{\beta})$ and $\boldsymbol{\beta}^*$. Summarizing the above two equations, we have

$$\sqrt{n}\mu(\boldsymbol{\beta}, \boldsymbol{x}_0) - \mu(\boldsymbol{\beta}^*, \boldsymbol{x}_0)) = \sqrt{n} \sum_{i \in \boldsymbol{\gamma}^*} ((\tilde{\nu}(\boldsymbol{\beta}_i) - \boldsymbol{\beta}_i^*)\mu_i(\boldsymbol{\beta}^*, \boldsymbol{x}_0) + o(1).$$

By Theorem 2.1, $\pi[\sqrt{n}(\tilde{\nu}(\boldsymbol{\beta}) - \boldsymbol{\beta}^*) \mid D_n] \rightsquigarrow N(0, \boldsymbol{V})$, where $\boldsymbol{V} = (v_{ij})$, and $v_{i,j} = E(h^{i,j}(\boldsymbol{\beta}^*))$ if $i, j \in \boldsymbol{\gamma}^*$ and 0 otherwise. Then we have $\pi[\sqrt{n}(\mu(\boldsymbol{\beta}, \boldsymbol{x}_0) - \mu(\boldsymbol{\beta}^*, \boldsymbol{x}_0)) \mid D_n] \rightsquigarrow N(0, \Sigma)$, where $\Sigma = \nabla_{\boldsymbol{\gamma}^*}\mu(\boldsymbol{\beta}^*, \boldsymbol{x}_0)^T H^{-1} \nabla_{\boldsymbol{\gamma}^*}\mu(\boldsymbol{\beta}^*, \boldsymbol{x}_0)$ and $H = E(-\nabla_{\boldsymbol{\gamma}^*}^2 l_n(\boldsymbol{\beta}^*))$.

## 3 Theory of Prior Annealing: Proof of Theorem 3.1

Our proof follows the proof of Theorem 2 in Chen et al. (2015). SGLD use the first order integrator (see Lemma 12 of Chen et al. (2015) for the detail). Then we have

$$\begin{aligned}
\mathbb{E}(\psi(\boldsymbol{\beta}^{(t+1)})) =&\psi(\boldsymbol{\beta}^{(t)}) + \epsilon_t \mathcal{L}_t \psi(\boldsymbol{\beta}^{(t)}) + O(\epsilon_t^2) \\
=&\psi(\boldsymbol{\beta}^{(t)}) + \epsilon_t(\mathcal{L}_t - \mathcal{L})\psi(\boldsymbol{\beta}^{(t)}) + \epsilon_t \mathcal{L}\psi(\boldsymbol{\beta}^{(t)}) + O(\epsilon_t^2).
\end{aligned}$$

Note that by Poisson equation, $\mathcal{L}\psi(\boldsymbol{\beta}) = \phi(\boldsymbol{\beta}) - \int \phi(\boldsymbol{\beta})\pi(\boldsymbol{\beta}|D_n, \eta^*, \sigma_{0,n}^*)d\boldsymbol{\beta}$. Taking expectation on both sides of the equation, summing over $t = 0, 1, \ldots, T-1$, and dividing $\epsilon T$ on both sides of

the equation, we have

$$\mathbb{E}\left(\frac{1}{T}\sum_{t=1}^{T-1}\phi(\boldsymbol{\beta}^{(t)}) - \int \phi(\boldsymbol{\beta})\pi(\boldsymbol{\beta}|D_n,\eta^*,\sigma_{0,n}^*)\right)$$

$$=\frac{1}{T\epsilon}(\mathbb{E}(\psi(\boldsymbol{\beta}^{(T)})) - \psi(\boldsymbol{\beta}^{(0)})) - \frac{1}{T}\sum_{t=0}^{T-1}\mathbb{E}(\delta_t\psi(\boldsymbol{\beta}^{(t)})) + O(\epsilon).$$

To characterize the order of $\delta_t = \mathcal{L}_t - \mathcal{L}$, we first study the difference of the drift term

$$\nabla\log(\pi(\boldsymbol{\beta}^{(t)}|D_{m,n}^{(t)},\eta^{(t)},\sigma_{0,n}^{(t)})) - \nabla\log(\pi(\boldsymbol{\beta}^{(t)}|D_n,\eta^*,\sigma_{0,n}^*))$$

$$=\sum_{i=1}^{n}\nabla\log(p_{\boldsymbol{\beta}^{(t)}}(\boldsymbol{x}_i,y_i)) - \frac{n}{m}\sum_{j=1}^{m}\nabla\log(p_{\boldsymbol{\beta}^{(t)}}(\boldsymbol{x}_{i_j},y_{i_j}))$$

$$+ \eta^{(t)}\nabla\log(\pi(\boldsymbol{\beta}^{(t)}|\lambda_n,\sigma_{0,n}^{(t)},\sigma_{1,n})) - \eta^*\nabla\log(\pi(\boldsymbol{\beta}^{(t)}|\lambda_n,\sigma_{0,n}^*,\sigma_{1,n})).$$

Use of the mini-batch data gives an unbiased estimator of the full gradient, i.e.

$$\mathbb{E}(\sum_{i=1}^{n}\nabla\log(p_{\boldsymbol{\beta}^{(t)}}(\boldsymbol{x}_i,y_i)) - \frac{n}{m}\sum_{j=1}^{m}\nabla\log(p_{\boldsymbol{\beta}^{(t)}}(\boldsymbol{x}_{i_j},y_{i_j}))) = 0.$$

For the prior part, let $p(\sigma)$ denote the density function of $N(0,\sigma)$. Then we have

$$\nabla\log(\pi(\boldsymbol{\beta}^{(t)}|\lambda_n,\sigma_{0,n}^{(t)},\sigma_{1,n}))$$

$$=-\frac{(1-\lambda_n)p(\sigma_{0,n}^{(t)})}{(1-\lambda_n)p(\sigma_{0,n}^{(t)}) + \lambda_n p(\sigma_{1,n})}\frac{\boldsymbol{\beta}^{(t)}}{\sigma_{0,n}^{(t)\,2}} - \frac{\lambda_n p(\sigma_{1,n})}{(1-\lambda_n)p(\sigma_{0,n}^{(t)}) + \lambda_n p(\sigma_{1,n})}\frac{\boldsymbol{\beta}^{(t)}}{\sigma_{1,n}^2},$$

and thus $\mathbb{E}|\nabla\log(\pi(\boldsymbol{\beta}^{(t)}|\lambda_n,\sigma_{0,n}^{(t)},\sigma_{1,n}))| \leq \frac{2\mathbb{E}|\boldsymbol{\beta}^{(t)}|}{\sigma_{0,n}^{*\,2}}$. By Assumption 5.2, we have

$$\mathbb{E}(|\eta^{(t)}\nabla\log(\pi(\boldsymbol{\beta}^{(t)}|\lambda_n,\sigma_{0,n}^{(t)},\sigma_{1,n})) - \eta^*\nabla\log(\pi(\boldsymbol{\beta}^{(t)}|\lambda_n,\sigma_{0,n}^*,\sigma_{1,n}))|)$$

$$=\mathbb{E}(|\eta^{(t)}\nabla\log(\pi(\boldsymbol{\beta}^{(t)}|\lambda_n,\sigma_{0,n}^{(t)},\sigma_{1,n})) - \eta^*\nabla\log(\pi(\boldsymbol{\beta}^{(t)}|\lambda_n,\sigma_{0,n}^{(t)},\sigma_{1,n}))|)$$

$$+ \mathbb{E}(|\eta^*\nabla\log(\pi(\boldsymbol{\beta}^{(t)}|\lambda_n,\sigma_{0,n}^{(t)},\sigma_{1,n})) - \eta^*\nabla\log(\pi(\boldsymbol{\beta}^{(t)}|\lambda_n,\sigma_{0,n}^*,\sigma_{1,n}))|)$$

$$\leq\frac{2M}{\sigma_{0,n}^{*\,2}}|\eta^{(t)} - \eta^*| + \eta^* M|\sigma_{0,n}^{(t)} - \sigma_{0,n}^*|.$$

By Assumption 5.1, $\mathbb{E}(\psi(\boldsymbol{\beta}^{(t)})) \leq \infty$. Then

$$\frac{1}{T}\sum_{t=0}^{T-1}\mathbb{E}(\delta_t\psi(\boldsymbol{\beta}^{(t)})) = O\left(\frac{1}{T}\sum_{t=0}^{T-1}(|\eta^{(t)} - \eta^*| + |\sigma_{0,n}^{(t)} - \sigma_{0,n}^*|)\right).$$

Note that by assumption 5.1, $|(\psi(\boldsymbol{\beta}^{(T)})) - \psi(\boldsymbol{\beta}^{(0)})|$ is bounded. Then

$$\mathbb{E}\left(\frac{1}{T}\sum_{t=1}^{T-1}\phi(X_t) - \int \phi(\boldsymbol{\beta})\pi(\boldsymbol{\beta}|D_n,\eta^*,\sigma_{0,n}^*)\right) = O\left(\frac{1}{T\epsilon} + \frac{\sum_{t=0}^{T-1}(|\eta^{(t)} - \eta^*| + |\sigma_{0,n}^{(t)} - \sigma_{0,n}^*|)}{T} + \epsilon\right).$$

## 4 Construct Confidence Interval

Theorem 2.2 implies that a faithful prediction interval can be constructed for the sparse neural network learned by the proposed algorithms. In practice, for a normal regression problem with noise $N(0,\sigma^2)$, to construct the prediction interval for a test point $\boldsymbol{x}_0$, the terms $\sigma^2$ and $\Sigma = \nabla_{\gamma^*}\mu(\boldsymbol{\beta}^*,\boldsymbol{x}_0)^T H^{-1}\nabla_{\gamma^*}\mu(\boldsymbol{\beta}^*,\boldsymbol{x}_0)$ in Theorem 2.2 need to be estimated from data. Let $D_n = (\boldsymbol{x}^{(i)},y^{(i)})_{i=1,\dots,n}$ be the training set and $\mu(\boldsymbol{\beta},\cdot)$ be the predictor of the network model with parameter $\boldsymbol{\beta}$. We can follow the following procedure to construct the prediction interval for the test point $\boldsymbol{x}_0$:

- Run algorithm 1, let $\hat{\boldsymbol{\beta}}$ be an estimation of the network parameter at the end of the algorithm and $\hat{\boldsymbol{\gamma}}$ be the correspoding network structure.

- Estimate $\sigma^2$ by

$$\hat{\sigma}^2 = \frac{1}{n}\sum_{i=1}^{n}(\mu(\hat{\boldsymbol{\beta}}, \boldsymbol{x}^{(i)}) - y^{(i)})^2.$$

- Estimate $\Sigma$ by

$$\hat{\Sigma} = \nabla_{\hat{\boldsymbol{\gamma}}}\mu(\hat{\boldsymbol{\beta}}, \boldsymbol{x}_0)^T(-\nabla_{\hat{\boldsymbol{\gamma}}}^2 l_n(\hat{\boldsymbol{\beta}}))^{-1}\nabla_{\hat{\boldsymbol{\gamma}}}\mu(\hat{\boldsymbol{\beta}}, \boldsymbol{x}_0).$$

- Construct the prediction interval as

$$\left(\mu(\hat{\boldsymbol{\beta}}, \boldsymbol{x}_0) - 1.96\sqrt{\frac{1}{n}\hat{\Sigma} + \hat{\sigma}^2}, \mu(\hat{\boldsymbol{\beta}}, \boldsymbol{x}_0) + 1.96\sqrt{\frac{1}{n}\hat{\Sigma} + \hat{\sigma}^2}\right).$$

Here, by the structure selection consistency (Lemma 2.2) and consistency of the MLE for the learnt structure Portnoy (1988), we replace $\boldsymbol{\beta}^*$ and $\boldsymbol{\gamma}^*$ in Theorem 2.2 by $\hat{\boldsymbol{\beta}}$ and $\hat{\boldsymbol{\gamma}}$.

If the dimension of the sparse network is still too high and the computation of $\hat{\Sigma}$ becomes prohibitive, the following Bayesian approach can be used to construct confidence intervals.

- Running SGMCMC algorithm to get a sequence of posterior samples: $\boldsymbol{\beta}^{(1)}, \ldots, \boldsymbol{\beta}^{(m)}$.

- Estimating $\sigma^2$ by $\hat{\sigma}^2 = \frac{1}{n}\sum_{i=1}^{n}(y^{(i)} - \mu^{(i)})^2$, where

$$\mu^{(i)} = \frac{1}{m}\sum_{j=1}^{m}\mu(\boldsymbol{\beta}^{(j)}, \boldsymbol{x}^{(i)}), i = 1, \ldots, n,$$

- Estimate the prediction mean by

$$\hat{\mu} = \frac{1}{m}\sum_{i=1}^{m}\mu(\boldsymbol{\beta}^{(i)}, \boldsymbol{x}_0).$$

- Estimate the prediction variance by

$$\hat{V} = \frac{1}{m}\sum_{i=1}^{m}(\mu(\boldsymbol{\beta}^{(i)}, \boldsymbol{x}_0) - \hat{\mu})^2 + \hat{\sigma}^2.$$

- Construct the prediction interval as

$$(\mu - 1.96\sqrt{V}, \mu + 1.96\sqrt{V}).$$

## 5 Prior Annealing

In this section, we give some graphical illustration of the prior annealing algorithm. In practice, the negative log-prior puts penalty on parameter weights. The mixture Gaussian prior behaves like a piecewise $L_2$ penalty with different weights on different regions. Figure 1 shows the shape of a negative log-mixture Gaussian prior. In step (iii) of Algorithm 1, the condition $\pi(\boldsymbol{\gamma}_i = 1|\boldsymbol{\beta}_i) > 0.5$ splits the parameters into two parts. For the $\boldsymbol{\beta}_i$'s with large magnitudes, the slab component $N(0, \sigma_{1,n}^2)$ plays the major role in the prior, imposing a small penalty on the parameter. For the $\boldsymbol{\beta}_i$'s with smaller magnitudes, the spike component $N(0, \sigma_{0,n}^2)$ plays the major role in the prior, imposing a large penalty on the parameters to push them toward zero in training.

Figure 2 shows the shape of negative log-prior and $\pi(\boldsymbol{\gamma}_i = 1|\boldsymbol{\beta}_i)$ for different choices of $\sigma_{0,n}^2$ and $\lambda_n$. As we can see from the plot, $\sigma_{0,n}^2$ plays the major role in determining the effect of the prior. Let $\alpha$ be the threshold in step (iii) of Algorithm 1 of the main body, i.e. the positive solution to $\pi(\boldsymbol{\gamma}_i = 1|\boldsymbol{\beta}_i) = 0.5$. In general, a smaller $\sigma_{0,n}^2$ will result in a smaller $\alpha$. If a very small $\sigma_{0,n}^2$ is used

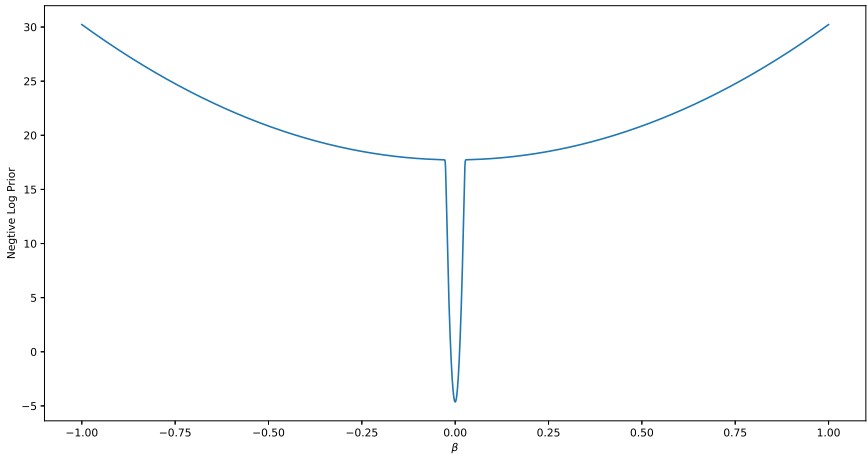

Figure 1: Negative logarithm of the mixture Gaussian prior.

in the prior from the beginning, then most of $\boldsymbol{\beta}_i$'s at initialization will have a magnitude larger than $\alpha$ and the slab component $N(0, \sigma_{1,n}^2)$ of the prior will dominate most parameters. As a result, it will be difficult to find the desired sparse structure. Following the proposed prior annealing procedure, we can start with a larger $\sigma_{0,n}^2$, i.e. a larger threshold $\alpha$ and a relatively smaller penalty for those $|\boldsymbol{\beta}_i| < \alpha$. As we gradually decrease the value of $\sigma_{0,n}^2$, $\alpha$ decreases, and the penalty imposed on the small weights increases and drives them toward zero. The prior annealing allows us to gradually sparsify the DNN and impose more and more penalties on the parameters close to 0.

## 6 Experimental Setups

### 6.1 Simulated examples

**Prior annealing**  We follow simple implementation of Algorithm given in section 3.1. We run SGHMC for $T = 80000$ iterations with constant learning rate $\epsilon_t = 0.001$, momentum $1 - \alpha = 0.9$ and subsample size $m = 500$. We set $\lambda_n = 1e-7, \sigma_{1,n}^2 = 1e-2, (\sigma_{0,n}^{init})^2 = 5e-5, (\sigma_{0,n}^{end})^2 = 1e-6$ and $T_1 = 5000, T_2 = 20000, T_3 = 60000$. We set temperature $\tau = 0.1$ for $t < T_3$ and for $t > T_3$, we gradually decrease temperature $\tau$ by $\tau = \frac{0.1}{t-T_3}$. After structure selection, the model is fine tuned for 40000 iterations. The number of iteration setup is the same as Sun et al. (2021).

**Other Methods**  Spinn, Dropout and DNN are trained with the same network structure using SGD with momentum. Same as our method, we use constant learning rate $0.001$, momentum $0.9$, subsample size $500$ and traing the model for $80000$ iterations. For Spinn, we use LASSO penalty and the regularization parameter is selected from $\{0.05, 0.06, \ldots, 0.15\}$ according to the performance on validation data set. For Dropout, the dropout rate is set to be $0.2$ for the first layer and $0.5$ for the other layers. Other baseline methods BART50, LASSO, SIS are implemented using R-package $randomForest$, $glmnet$, $BART$ and $SIS$ respectively with default parameters.

### 6.2 CIFAR10

We follow the standard training procedure as in Lin et al. (2020), i.e. we train the model with SGHMC for $T = 300$ epochs, with initial learning rate $\epsilon_0 = 0.1$, momentum $1 - \alpha = 0.9$, temperature $\tau = 0.001$, mini-batch size $m = 128$. The learning rate is divided by 10 at 150th and 225th epoch. We follow the implementation given in section 3.1 and use $T_1 = 150, T_2 = 200, T_3 = 225$, where $T_i$s are number of epochs. We set temperature $\tau = 0.01$ for $t < T_3$ and gradually decrease $\tau$ by $\tau = \frac{0.01}{t-T_3}$ for $t > T_3$. We set $\sigma_{1,n}^2 = 0.04$ and $(\sigma_{0,n}^{init})^2 = 10 \times (\sigma_{0,n}^{end})^2$ and use different $\sigma_{0,n}^{end}, \lambda_n$ for different network size and target sparsity level. The detailed settings are given below:

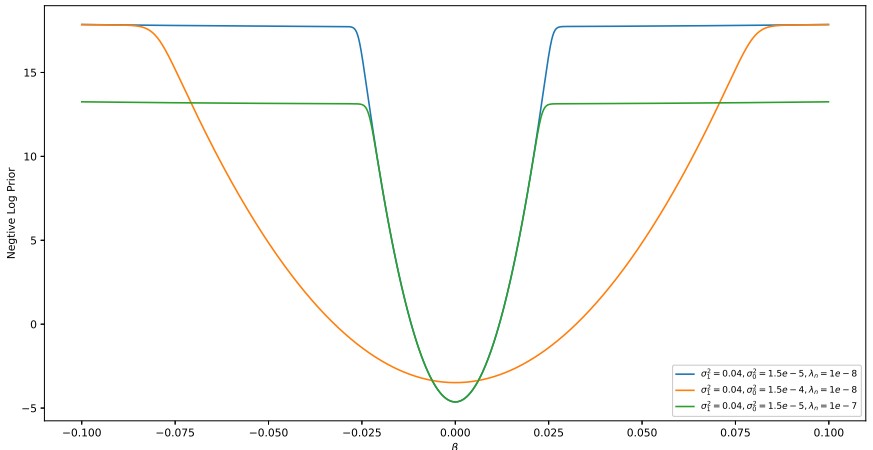

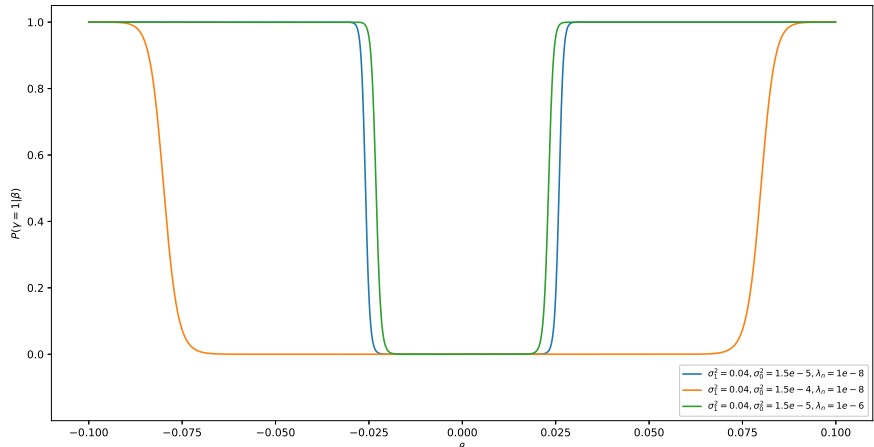

Figure 2: Negative log-prior and $\pi(\boldsymbol{\gamma} = 1|\boldsymbol{\beta})$ for different choices of $\sigma_{0,n}^2$ and $\lambda_n$.

- ResNet20 with target sparsity level 20%: $(\sigma_{0,n}^{end})^2 = 1.5e - 5, \lambda_n = 1e - 8$

- ResNet20 with target sparsity level 10%: $(\sigma_{0,n}^{end})^2 = 6e - 5, \lambda_n = 1e - 9$

- ResNet32 with target sparsity level 10%: $(\sigma_{0,n}^{end})^2 = 3e - 5, \lambda_n = 2e - 9$

- ResNet32 with target sparsity level 5%: $(\sigma_{0,n}^{end})^2 = 1e - 4, \lambda_n = 2e - 8$