# OpenReview forum: "Sparse Deep Learning: A New Framework Immune to Local Traps and Miscalibration"
_NeurIPS.cc/2021/Conference — NeurIPS 2021 Poster_

### Official Review · Reviewer_PeZg · 2021-07-15

**Rating:** 7
**Confidence:** 3

**Summary:**

1. Theoretical contributions: the authors extend the theory established by Sun et al. [2021] and study the asymptotic behaviors of sparse deep learning.
2. Methodological contributions: the authors propose prior annealing algorithms, one from the frequentist perspective and one from the Bayesian perspective, to implement the corresponding sparse neural nets. They show the performance of the prior annealing algorithms on one synthetic example and the CIFAR10 datasets.

**Limitations And Societal Impact:**

I didn’t spot a discussion on limitations. Maybe the authors can discuss based on the reviews and provide the future direction of their work.

**Main Review:**

The authors inherit the sparse DNN framework used in Sun et al. [2021], and they investigate the asymptotic normality of connection weights and predictions.  The theory established in the paper are around the true DNN model $\mu^*(\beta^*, \gamma^*, x)$, which is a data-dependent $L^2$ projection of $\mu^*(x)$ onto a pre-specified neural network space $\mathcal{G}_n$. My intuition is that one would need prior knowledge on what an adequate network structure looks like, and do the pruning based on that, like the CIFAR10 example. It appears more as a refinement to already mature neural net models.
And it is not clear how one can relate the theory to asymptotic behavior around $\mu^*(x)$ since that will depend on the approximation error. Maybe the authors can elaborate more on that. In addition, how the noise $\sigma^2$ could impact the model performance, is it assumed known?

For the two prior annealing algorithms, the results from the experiments look decent. But it is somewhat disconnected from the asymptotic theory and it is not obvious how to assess the uncertainty in prediction. It is not clear how the prediction intervals in Figure 1 are constructed, it seems like a variant of bootstrap.

Some minor comments:
1. The notation for log-likelihood function $l_n(\beta)$ sometimes appear as $nl_n(\beta)$. Line 126-127, the expression is not properly placed. Overall, the notations in the paper are really heavy and section 2.1 is directly borrowed from Sun et al. [2021].

2. For the references, some conferences as NeurIPS and ICML have inconsistent formats.


**Time Spent Reviewing:**

6

---

> ### Author Response · Authors · 2021-08-10
> **Explanation of theory and confidence interval procedure**
>
> Thanks for your comments and suggestions.
>
> Our framework start with over-parametrized neural network. The neural network space ${\cal G}_n$ should be large enough to contain neural network that provide adequate approximation of the underlying function $\mu^*(x)$. We follows the assumptions for the neural network space in [1], which is compatible with several approximation theory, e.g. [2] for functions represented by affine system, [3] for $\alpha$-H{\"o}lder smooth function. As discussed in [1], the framework used in the paper allow us to incorporate different approximation theory and represent the approximation error as a function of $n$. As the number of data increase, the network size is allow to increase and the approximation error will tends to 0. The posterior consistency ensures that the true DNN model can be identified, which will be an consistent estimate of the true function $\mu^*(x)$.
>
> In practice, to construct confidence intervals, the noise $\sigma^2$ will be estimated from data. Let $\hat{\beta}$ be an estimation of the network parameter at the end of the algorithm and $\hat{\gamma}$ be the correspoding network structure. Let $D_{n}=(x^{(i)},y^{(i)})  i=1,...,n$ be the training set and $\mu(\hat{\beta}, x^{(i)})$ be the prediction of the network model. We will estimate $\sigma^2$ by
> $$\hat{\sigma}^2 = \frac{1}{n} \sum_{i=1}^n ( \mu(\hat{\beta}, x^{(i)}) - y^{(i)})^2.$$
> To construct the confidence interval for a test point $z$, according to theorem 2.2, we estimate $\Sigma$ by
> $$
> \hat{\Sigma} =  \nabla_{\hat{\gamma}}\mu(\hat{\beta}, z)^{T}(-\nabla^2_{\hat{\gamma}} l_n(\hat{\beta}))^{-1}\nabla_{\hat{\gamma}}\mu(\hat{\beta}, z)
> $$
> Here, by the structure selection consistency(lemma 2.2) and consistency of the MLE for the learnt structure [4], we replace $\beta^*$ and $\gamma^*$ in theorem 2.2 by $\hat{\beta}$ and $\hat{\gamma}$. When the selected structure is still too large and the calculation of $\hat{\Sigma}$ become computational prohibitive, we can use the Bayesian procedure discussed in the paper to estimate $\hat{\Sigma}$.
>
> The confidence interval is given by
> $$
> \left(\mu(\hat{\beta}, z) - 1.96\sqrt{\frac{1}{n}\hat{\Sigma} + \hat{\sigma}^2}, \mu(\hat{\beta}, z) + 1.96\sqrt{\frac{1}{n}\hat{\Sigma} + \hat{\sigma}^2}\right)
> $$
>
> The notation will be fixed in the revision.
>
>
>
> ## Reference
> [1] Y. Sun, Q. Song, and F. Liang. Consistent sparse deep learning: Theory and computation.Journal of the American Statistical Association, page in press, 2021
>
> [2] Helmut Bolcskei, Philipp Grohs, Gitta Kutyniok, and Philipp Petersen. Optimal approximation with sparsely connected deep neural networks.SIAM Journal on Mathematics of Data Science, 1(1):8-45, 2019
>
> [3] Johannes Schmidt-Hieber. Nonparametric regression using deep neural networks with relu activation function. arXiv:1708.06633, 2017.
>
> [4] S. Portnoy. Asymptotic behavior of likelihood methods for exponential families when the number of parameters tend to infinity.The Annals of Statistics, 16(1):356–366, 1988

---

### Official Review · Reviewer_f5t3 · 2021-07-16

**Rating:** 7
**Confidence:** 2

**Summary:**

The submission addresses the issue of sparsifying trained deep neural networks in an optimal way, preserving performance and improving calibration. It builds on the formalism and results of Sun et al. (2021), proves additional results and proposes two prior-annealing algorithms, which they demonstrate on synthetic and real data, comparing favorably to existing sparsifying algorithms.

**Ethical Concerns:**

Minor point concerning citing existing assets: the SVHN dataset is mentioned and should be properly cited.

**Limitations And Societal Impact:**

The authors did mostly address the limitations of their work in the text.
Potential societal impact were not discussed, but this is mostly theoretical work, so it seems fine to me.

**Main Review:**

Originality
----------------
This submission relies heavily on work by Sun et al. (2021), which is properly acknowledged, and expands on that work. The two prior-annealing algorithms seem novel, and the real-data, larger-scale evaluation (ResNets on CIFAR-10, following the set up of Lin et al., 2020 for meaningful comparison) make that work more accessible to current deep learning practitioners.

Quality
----------
To the best of my understanding, the theory is sound, and the theoretical part of the paper feels complete. The claims of performance and calibration against other sparsity-inducing methods are well supported by experiments, and conclusions drawn by the authors are faithful to the experimental results.

Some of the broader claims may be over-stated, though ("how to make statistical inference with deep neural networks", "tamed the powerful deep neural networks", l.56-57), especially the ones that are not restricted to _sparse_ deep networks.
In order to support those claims, it would be better to have, for instance:
- comparisons of test accuracy against non-pruned network (in Table 2)
- comparisons of the measures of Table 3 (NLL, JS-Distance, ECE) against values obtained by algorithms aimed at improving calibration (e.g., temperature scaling, mixup training, focal loss training...) for regular, dense DNNs, rather than only considering sparsifying algorithms.

Clarity
----------
Considering the complexity of the theoretical work, the article makes a good job at explaining the issues and results informally, giving a good intuition. The overall organization makes sense, and I appreciate the division between main paper and appendix.

I believe a reader already well familiar with Sun et al. (2021) could implement the additional procedures described in this submission and reproduce the results.

Significance
-----------------
 The perspective of taking a trained, well-performing DNN and turning it into a sparse, well-calibrated model, in a principled way, and with minimal loss of performance, is quite appealing, and I expect practitioners will want to use this method.
This submission adds additional theoretical guarantees compared to earlier work, making this method more solid.



**Time Spent Reviewing:**

9

---

> ### Author Response · Authors · 2021-08-10
> **Additional comparison with dense model**
>
> Thanks for your comments and suggestions.
>
> We will add comparison with non-pruned network. For our simulated example, training a dense network will over-fit the data. We use the same training setup as our method and train the model with 10 data set. The dense model has MSFE $1.1701e-5(1.1542e-6)$ and MSPE $16.9226(0.3230)$. Even with early stopping using an additional validation data set, an Dense DNN model has MSFE $6.4498(0.4132)$ and MSPE $10.0025(0.1598) $.
>
> The CIFAR10 example is an application of our model to residual network pruning. For this network pruning example with low pruning ratio, the sparse network typically has worse test accuracy compared to dense model (see e.g. [1]). The main purpose is to reduce the size of the network while maintaining similar accuracy. We will add comparison of our method with dense model. Some results are given in the following table(smaller values of NLL and ECE, larger values of JS-distance imply better model calibration). The dense network is trained with same optimization set up as our method. Although the sparse network has worse accuracy, it outperforms the dense network in terms of ECE and JS-Distance, which indicates that in addition to reduce the model size, finding sparse network structure is also one way toward better calibration. Our framework can be combined with other method toward improving calibration. e.g. we can also incorporate temperature scaling in the softmax calculation.
>
> | Method   |      Model      | Pruning Ratio  | Test Accuracy | NLL | JS-Distance | ECE |
> |----------|:-------------:|:-------------:|:-------------:|:-------------:|:-------------:|:-------------:|
> | BNN\_average | ResNet20 | {9.88\%(0.02\%)} | 91.65(0.08) | 0.2528(0.0029) | 9.9641(0.3069) | 0.0113(0.0010) |
> | BNN\_anneal  | ResNet20 | {9.95\%(0.03\%)}  | 91.28(0.11) | 0.2618(0.0037) | 10.1251(0.1797) | 0.0175(0.0011) |
> DNN\_dense  | ResNet20 | 100\%  | 92.93(0.04) | 0.2276(0.0021)| 7.9118(0.9316)  | 0.02627(0.0005）|
> | BNN\_average | ResNet32 | 9.99\%(0.08\%) | {93.12(0.09)} | 0.2116(0.0012) | 9.4549(0.5456) | 0.0132(0.0001)  |
> | BNN\_anneal  | ResNet32 | 9.97\%(0.03\%) | {92.63(0.09)} | 0.2218(0.0013) | 8.5447(0.1393) | 0.0192(0.0009) |
> | DNN\_dense | ResNet32 | 100\%  | 93.76(0.02) | 0.2042(0.0017) | 6.7699(0.5253)  | 0.02613(0.00029) |
>
> ## Reference
> [1] Song Han, Jeff Pool, John Tran, and William Dally.  Learning both weights and connections for efficient neural network.   In Advances in neural information processing systems,  pages 1135–1143, 2015

---

> > ### Comment · Reviewer_f5t3 · 2021-08-30
> > **About comparisons with dense models**
> >
> > I think that maybe the authors misunderstood my point about comparing with dense models.
> > As I understand it, this work comes from a line of research centered on _sparse_ neural networks, where the goal is usually to find a trade-off between sparsity and accuracy. The experiments performed reflect that setting, which is fine, and I think they accurately represent the benefits of the proposed method, compared to others in the literature.
> >
> > However, even with the additional results reported, I don't think they are enough to support the broader claims that this method should be preferred for DNN training in general. L 55-57, in particular, "we have tamed the powerful deep neural networks into the framework of statistical modeling" implies this method is superior to the existing methods for training even over-parameterized networks in general.
> > According to your latest experiments, though, this is not the case, and there are trade-offs, regarding accuracy in particular, and there may be others if you would compare with methods (other than sparsity) aiming at improving calibration.
> >
> > This is not to say there are not cases where not worth it, even when sparsity is not the primary goal. And I appreciate the additional results. But I think the claims of general superiority should be toned down, and these trade-offs discussed if non-sparse DNNs (or just "DNNs") are mentioned.

---

> > > ### Author Response · Authors · 2021-08-31
> > > **Tone down general claim**
> > >
> > > Thanks for your reply.
> > >
> > > Our intention is to emphasis that we can make uncertainty quantification for deep learning under the sparse network and provide theoretical justification for constructing faithful confidence region, while the theoretical property of some methods aiming at improving calibration(e.g. temperature scaling) is not clear. We agree with you that in practice, there are still trade-offs regarding sparsity and accuracy, calibration, etc.
> > >
> > > As suggested, we will tone down the general claim when revising the paper.

---

> ### Comment · Reviewer_f5t3 · 2021-08-31
> **Possible typo?**
>
> I just realized the MSFE for Dropout (Table 1) is 1.104, which is the smallest, and quite different from the MSPE (13.183), when they usually have the same order of magnitude. Moreover, the corresponding number in Sun et al. (2021) is 10.491, which is close to 10x higher.
>
> Is it just a typo, or do you have an explanation for that discrepancy?

---

> > ### Author Response · Authors · 2021-08-31
> > **Discrepancy of MSFE and MSPE**
> >
> > Thanks for your comments.
> >
> > This is not a typo. For the simulation experiments, Sun et al. (2021)[1] used a network of size 2000-6-4-3-1. In this paper, we intentionally use a large network of size 2000-10000-100-10-1 to satisfy the pyramidal shape condition in [2]. For this example, the results in [1] indicate that for a smaller network, dropout can prevent over-fitting, but without selecting correct variables, the model cannot make accurate prediction. While our results indicate that for an extremely large network, dropout cannot prevent over-fitting.
> >
> > ## Reference
> > [1] Y. Sun, Q. Song, and F. Liang. Consistent sparse deep learning: Theory and computation.Journal of the American Statistical Association, page in press, 2021
> >
> > [2] Quynh Nguyen and Matthias Hein.  The loss surface of deep and wide neural networks.  In ICML, 2017

---

### Official Review · Reviewer_5FUc · 2021-07-16

**Rating:** 5
**Confidence:** 2

**Summary:**

Overall, the work presents an interesting direction into NN pruning. However, the paper is quite hard to follow and overly complex in notation, with little effort to assist the reader. Further, the paper severely lacks useful devices, such as figures or in-depth description, to explain the methods to readers, making the work challenging to follow.  Last, some of the experimental evaluation is somewhat questionable, as specified in the details comments.  Though the direction seems useful, additional work is needed in communicating the results and in experiments in order to warrant acceptance.


**Main Review:**

Per Section Comments
———————————

Section 2.1:
- The paper is reasonable to follow up until the statement of assumptions (around line 106). However, at this point, why are these assumptions reasonable? It’s tricky to see what some of these assumptions mean (i.e., A.2.1 in particular) and why it might be reasonable to make other assumptions (i.e., A.1, this is rarely the case in practice, A.2.2 why we expect the polynomial bound to hold).  Even if it is the case that some of these assumptions may only be plausible, for the sake of the analysis, more context at this point would be highly useful for the reader.

Section 3.1
- Gradually decreasing the temperature: How is this specified?
- It would be highly useful to provide some more intuition around algorithm 1.  Where does the constraint in (iii) come from, for example. Is \sigma_0^init define anywhere?
- A graphical figure here to explain the algorithm in more detail would be highly useful.

Experiments:
- Investigating the results provided in section 4.2, it seems like the proposed methods yield slightly test accuracy. Yet, the pruning ratio is slightly lower as well for these networks. Because the difference in accuracy is so slight and the pruning ratio changes slightly as well, can you account for these changes in evaluation? Looking over the results, it seems hard to justify that the method is better than baselines, given this dynamic, as the results are currently written.
- Further, the proposed method (if I’m not mistaken) uses additional fine-tuning after the variable selection is applied (i.e., IV of alg. 1).  Do the other variable selection methods do additional fine-tuning? Some discussion about this would be good, because it seems difficult to compare this method (which allows further fine-tuning) with methods that do not.


**Time Spent Reviewing:**

3

---

> ### Author Response · Authors · 2021-08-10
> **Explanation of assumption, algorithm and additional results**
>
> Thanks for your comments and suggestions
>
> Section 2.1
>
> The assumptions are taken from [1], detailed explanations of these assumptions can be found in Section 2.2 of [1]. In the revision, we will add remarks on the assumptions to the supplementary material. Assumption A.1 is a widely used assumption for posterior consistency(see [2],[3]). It essentially requires ${ x}$ to be bounded by some constant, the use of $[-1, 1]$ is mainly for convenience. In practice, bounded data can be normalized to satisfy the assumption, e.g. image data are typically normalized before training. Assumption A.2.1 restricts the sparsity of the model, the condition $r_nH_n\log n +r_n\log\overline L+s_n\log p_n\leq C_0n^{1-\varepsilon}$ controls the number of connections $r_n$ and input variable $p_n$, gives an upper bound of their order. The polynomial bound in A.2.2 coincide with the approximation theory of sparse neural network in [4], neural network satisfying the polynomial bound and assumption A.2.1 can still be used to approximate functions that can be represented by affine system [4].
>
> Section 3.1
>
> In all our experiments, we linearly decrease the temperature $\tau$, for our simulated example, we set $\tau = \frac{0.1}{t -T_3}$, for CIFAR10 example, we set $\tau = \frac{0.01}{t - T_3}$. Those schedule has been specified in supplementary material(line 99, 115). The condition in (iii) can be obtained by solving the conditional distribution $\pi(\gamma_i = 1 | \beta_i) > 0.5$ based on the mixture normal prior. It is also used in [1] to select structure of network. $\sigma_{0,n}^{init}$ is the starting value of $\sigma_{0,n}$ during the prior annealing, which can be viewed as a hyper-parameter for the algorithm. For a given $\sigma_{0,n}$, the condition in step (iii), i.e.  $\pi(\gamma_i = 1 | \beta_i) > 0.5$ split the parameters into two parts, for $\beta_i$ with larger magnitude, $N(0, \sigma_{1,n}^2)$ plays major role in the prior, for $\beta_i$ with smaller magnitude, $N(0, \sigma_{0,n}^2)$ plays major role in the prior. As we decrease the value of $\sigma_{0,n}^2$, the value of the threshold in step (iii) decrease, and for $\beta_i$ less than the threshold, the penalty put by the prior increase(for those $\beta_i$,the prior is dominated by $N(0, \sigma_{0,n}^2)$). In practice, the prior annealing is essentially gradually sparsify the model and put more and more penalty on parameters close to 0. We will add graphical figure to explain the change of prior shape as well as the annealing schedule for some hyper-parameters.
>
> Experiments:
>
> For CIFAR10 experiments, for a same model, at low pruning ratio, decreasing the pruning ratio will typically decrease test accuracy as well [5]. Our comparison with DPF etc. favors the baseline model. We didn't have the exact pruning ratio because our threshold is calculated in step (iii) of the algorithm.
>
> BNN\_BIC also use fine tuning, we follow the same set up as [1], the model was retrained for only one epoch for refining the
> nonzero weights.
>
> In the revision, we will add results without fine tuning and with the same pruning ratio as the DPF method(pruning certain percentage of the weights with smallest magnitude). Some results that compare the new result with DPF are summarized in the following table. With the same pruning ratio and without fine tuning. Our model still outperform DPF
>
>
> | Method   |      Model      | Pruning Ratio  | Test Accuracy | NLL | JS-Distance | ECE |
> |----------|:-------------:|:-------------:|:-------------:|:-------------:|:-------------:|:-------------:|
> | BNN\_anneal  | ResNet20 | 20\%  | 92.21(0.02) | 0.2422(0.0019) | 8.8681(0.3173) | 0.0243(0.0004) |
>  | DPF  | ResNet20 | 20\%  | 92.17(0.21) |  0.2874(0.0029) | 7.7329(0.1400) | 0.0391(0.0001) |
> | BNN\_anneal  | ResNet20 | 10\%  | 90.99(0.02) | 0.2670(0.0030) | 7.7256(0.4599)  | 0.0195(0.0010)   |
> | DPF  | ResNet20 | 10\%  | 90.88(0.07) |  0.2833(0.0004)| 7.5712(0.4466)  | 0.0294(0.0009)  |
> | BNN\_anneal  | ResNet32 | 10\%  | 92.71(0.06)  | 0.2204(0.0015) | 8.0318(0.4866) | 0.0195(0.0007)   |
> | DPF  | ResNet32 | 10\%  | 92.42(0.18) |  0.2677(0.0041)| 7.8693(0.1840) | 0.0364(0.0015)  |
> | BNN\_anneal  | ResNet32 | 5\%  | 91.09(0.08) | 0.2749(0.0030) | 9.3348(0.7499) | 0.0191(0.0006)   |
> | DPF  | ResNet32 | 5\%  | 90.94(0.35) | 0.2921(0.0067)| 6.3990(0.8384)  | 0.0276(0.0019)  |
>
>
>
>
> ## Reference
> [1] Y. Sun, Q. Song, and F. Liang. Consistent sparse deep learning: Theory and computation.Journal of the American Statistical Association, page in press, 2021
>
> [2] Nicholas G. Polson and Veronika Rockova. Posterior concentration for sparse deep learning. In Proceedings of the 32nd International Conference on Neural Information Processing Systems
>
> [3] Wenxin Jiang et al. Bayesian variable selection for high dimensional generalized linear models: convergence rates of the fitted densities.The Annals of Statistics, 35(4):1487-1511, 2007
>
> [4] Helmut Bolcskei, Philipp Grohs, Gitta Kutyniok, and Philipp Petersen. Optimal approximation with sparsely connected deep neural networks.SIAM Journal on Mathematics of Data Science, 1(1):8-45, 2019
>
> [5] Song Han, Jeff Pool, John Tran, and William Dally.  Learning both weights and connections208for efficient neural network.   InAdvances in neural information processing systems,  pages 1135-1143, 2015

---

### Official Review · Reviewer_fjLT · 2021-07-17

**Rating:** 5
**Confidence:** 3

**Summary:**

The paper proposes an approach to learning sparse deep neural networks (neural networks where most of the weights are set to zero) using annealing: starting by training a large overparameterized neural network to fit the training data perfectly, then adding weight penalties according to a prespecified schedule. A theoretical argument is provided to argue why this is reasonable, with two new theory proofs provided which extend previous work presented in Sun et al. 2021. Experiments on synthetic data show that the proposed algorithms perform feature selection better than other existing algorithms. Experiments on CIFAR10 show the algorithms are competitive in terms of the accuracy vs. pruning tradeoff.

**Limitations And Societal Impact:**

The authors adequately address this.

**Main Review:**

This paper is mostly clear and well-written, aside from some typos. However, its contribution is unclear to me. I'll address each contribution in turn:

Theory: Proofs are provided for the asymptotic normality of the connection weights and asymptotic normality of the prediction. These appear to be novel contributions that build upon the previous work of Sun et al. 2021. I did not check the proofs carefully, but they appear to be non-trivial and novel contributions. However, I was unable to see their significance; the implications for the algorithms were unclear.

Algorithms: Two algorithms for training sparse DNNs are proposed. These algorithms seem reasonable, but I was left wondering about the theoretical and practical value. Line 230 says, “the MCMC algorithm is likely to hit the region around the optimum of the target distribution,” but if nothing concrete can be stated what is the advantage of starting from the global optimum and then annealing? Practically, this annealing procedure is more computationally intensive than training with standard regularization methods, and we are left with an empirical question of whether the annealing is more computationally-efficient than other annealing methods.

Experiments: the experiments show that the annealing approach results in sparse networks that have competitive accuracy and calibration to other methods. However, I have concerns about the fairness of these experiments. As mentioned in Guo 2017, miscalibration is a result of overfitting and can largely be avoided by early stopping on the validation set NLL (not 01-loss). However, early stopping was not mentioned, and in the supplementary materials the authors say the benchmark comparison models were trained for a fixed number of epochs. In addition, the hyperparameters (lambda, sigma_0^end) for the annealing algorithms appear to be chosen in order to give good results, reducing my confidence in the fairness of these comparisons. Overall, I'm not convinced that this is a promising approach for training sparse DNNs in practice, if that was the goal.



**Time Spent Reviewing:**

6

---

> ### Author Response · Authors · 2021-08-10
> **Explanation of theory, algorithm and additional results**
>
> Thanks for your comments and suggestions.
>
> Theory: Our theory established the asymptotic normality of connection weights and prediction, which implies that a faithful prediction interval can be constructed for the sparse neural network learned by the proposed algorithms. Numerical evidence has been shown in Figure 1 of the paper.
>
> Algorithm: For over-parametrized neural network with mixture of normal prior, in the beginning of optimization, the prior part may dominate the likelihood part, making the energy landscape highly non-convex and hard to find the desired optimal. Intuitively, starting from the global optimum allow the algorithm to easily find the optimal region of the distribution at each stage of prior annealing. In practice, if a very small value $\sigma_{0,n}^2$ is used in the prior from the beginning, then for most of the network parameters at initialization, the $N(0, \sigma_{1,n}^2)$ part in prior will dominate the prior. Then it will be hard to find the desired sparse structure. Compared to standard annealing method, our prior annealing phase allow the algorithm to gradually find desired sparse structure.
>
> The computation complexity of the proposed algorithm for each iteration is similar to standard regularization method or annealing method. Step (i) and (iv) of the algorithm could be very short. As being shown in the experiments, our method could achieve better result using same iterations as regularization method such as [1]. We will also add training loss vs epoch plot in the revision.
>
> Experiments: CIFAR10 example is an application of our method to network pruning. For CIFAR10 experiment with ResNet and decreasing learning rate schedule, early stopping time will be entangled with learning rate schedule. Running fixed number of epochs is the standard training setup and the test NLL and accuracy for sparse network is rather stable after 225 epochs(when the smallest learning rate 0.001 is used, see e.g. Figure 4 in [1]).
>
> Even with early stopping, the baseline method still performs worse than our method. E.g., for DPF, in order to get early stopping result, we use code in https://github.com/INCHEON-CHO/Dynamic_Model_Pruning_with_Feedback to reproduce the result. For ResNet20 with 10\% target sparsity level, we run the experiment 3 times and use the model with the best test NLL loss for last 75 epoch to do prediction(in order to avoid splitting validation data set and making the training data different and comparison unfair, we directly use the test set for early stopping, which in favors the baseline method), we get test accuracy $90.93(0.12)$, MNLL $0.2763(0.0030)$, JS-Distance $8.8186(0.3219)$, ECE $0.02663(0.00158)$, which is still worse than our method.
>
> The test accuracy results of baseline methods for CIFAR10 example used in the paper are taken from [2] and [1] where the models are also properly tuned.
>
> ## Reference
> [1] Y. Sun, Q. Song, and F. Liang. Consistent sparse deep learning: Theory and computation.Journal of the American Statistical Association, page in press, 2021
>
> [2] Tao Lin, Sebastian U. Stich, Luis Barba, Daniil Dmitriev, and Martin Jaggi.  Dynamic model pruning with feedback. In International Conference on Learning Representations, 2020

---

### Decision · Program_Chairs · 2021-09-27

**Decision:**

Accept (Poster)

**Comment:**

This paper proposed a framework for sparse deep learning, with the primary goals as addressing mis-calibration problems. The paper contains both theoretical analysis as well as some experiments to demonstrate the theoretical claims on Bayesian ResNets.

This paper sits at the intersection of network pruning and uncertainty calibration so it might benefit both communities. However, the paper heavily assumes prior knowledge presented in Sun et al. (2021) which is a very recent JASA paper. I would expect this paper to be very difficult to read for deep learning practitioners interested in network pruning.

The paper stated that "This work, together with Sun et al. (2021), has built a solid theoretical foundation for sparse deep learning". Therefore in my view, acceptance decision critically relies on how significant the contribution of this paper is on top of Sun et al. (2021).
- From my understanding after a brief read, the key theoretical result in this paper is the asymptotic consistency/normally estimate for the defined "ground truth" parameters.
- While Sun et al. (2021) established posterior consistency and structure identification consistency, that is regarding the estimation of the data distribution, rather than the underlying parameters.
- Therefore I think the two paper's theoretical results are different aspects of the same framework. This makes me think that in some sense this is a project split into two publications?

Experimentally, the proposed approach and the approach of Sun et al. (2021) perform quite close in many of the experiments.
- This also raises a practical question in that, to what extent it is important to ensure "consistency in parameter estimation"?
- It is even more confusing that the "ground truth" parameters are defined using "universal approximation of neural network", so I would think "consistency in function estimation" might be more important here, which is already discussed in Sun et al. (2021) to certain extent.